# MONGOOSE: A LEARNABLE LSH FRAMEWORK FOR EFFICIENT NEURAL NETWORK TRAINING

**Beidi Chen**[1]**, Zichang Liu**[2]**, Binghui Peng**[3]**, Zhaozhuo Xu**[2]**, Jonathan Lingjie Li**[1]**, Tri Dao**[1]**,
Zhao Song**[4]**, Anshumali Shrivastava**[2]**, Christopher Ré**[1]
[1] Stanford University, [2] Rice University, [3] Columbia University, [4] Princeton University
`{beidic,jlli,trid,chrismre}@stanford.edu, bp2601@columbia.edu`
`{zl71,zx22,anshumali}@rice.edu, zhaos@princeton.edu`

## ABSTRACT

Recent advances by practitioners in the deep learning community have breathed new life into Locality Sensitive Hashing (LSH), using it to reduce memory and time bottlenecks in neural network (NN) training. However, while LSH has sub-linear guarantees for approximate near-neighbor search in theory, it is known to have inefficient query time in practice due to its use of random hash functions. Moreover, when model parameters are changing, LSH suffers from update overhead. This work is motivated by an observation that model parameters evolve slowly, such that the changes do not always require an LSH update to maintain performance. This phenomenon points to the potential for a reduction in update time and allows for a modified learnable version of data-dependent LSH to improve query time at a low cost. We use the above insights to build MONGOOSE, an end-to-end LSH framework for efficient NN training. In particular, MONGOOSE is equipped with a scheduling algorithm to adaptively perform LSH updates with provable guarantees and learnable hash functions to improve query efficiency. Empirically, we validate MONGOOSE on large-scale deep learning models for recommendation systems and language modeling. We find that it achieves up to 8% better accuracy compared to previous LSH approaches, with $6.5\times$ speed-up and $6\times$ reduction in memory usage.

## 1 INTRODUCTION

Locality Sensitive Hashing (LSH) has been adapted to address the computational and memory bottlenecks of large-scale neural network (NN) training in natural language processing (Chandar et al., 2016; Rae et al., 2016; Kitaev et al., 2020), computer vision (Chen et al., 2015) and recommendation systems (Spring & Shrivastava, 2017; Chen et al., 2020). Specifically, giant matrix multiplications in linear layers preceding a softmax can be approximated using nearest neighbor search (NNS) techniques, which often rely on LSH. However, LSH methods used in NNs are inefficient. Although LSH achieves sub-linear query time in theory, it is known to suffer from high query and pre-processing (update) overhead in practice (Erik et al., 2018). In the setting of NN training, where data points for LSH are model parameters, this overhead is exacerbated by a high number of updates due to constantly evolving model parameters.

The most established solution for reducing LSH query overhead is data-dependent or learning-based hashing, which uses adaptive hash functions to optimize the LSH bucket distribution for each dataset (Andoni & Razenshteyn, 2015; Dong et al., 2019). These methods reduce query time by incurring a one-time offline cost to learn useful data input patterns in a preprocessing step. The learning techniques are often computationally complex, but can lead to a net reduction in overall query time. However, in NN training, the expensive preprocessing procedure has to be repeated each time the parameters are updated. Naïvely applying these techniques would increase the LSH update overhead rather than reduce it. A more appropriate data-dependent LSH framework would ideally (i) have a deeper understanding of the training dynamics of model parameters, a setting in which LSH has not been well-studied, (ii) be able to perform low-cost updates to account for evolving parameters, and (iii) have better query time while accurately approximating matrix multiplication.

We argue that it is unnecessary to view evolving model parameters as streaming data and not every parameter update requires an LSH update. In fact, LSH updates are necessary only when the NN gra-

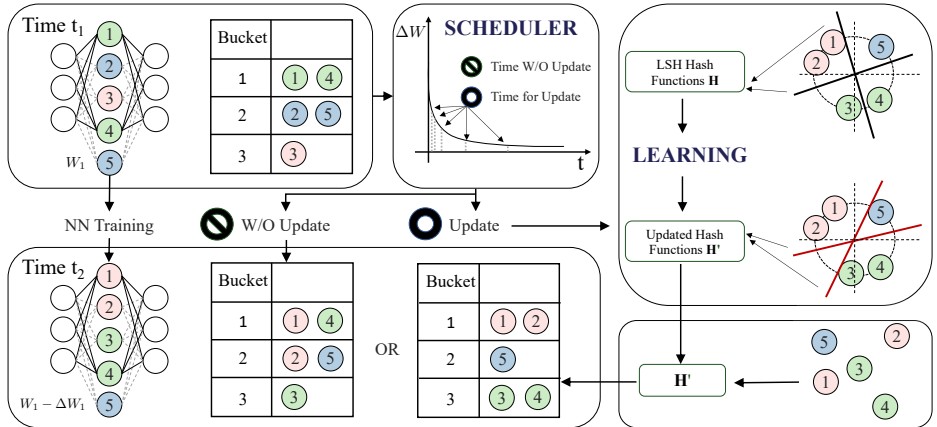

Figure 1: MONGOOSE workflow - For each layer in the NN, besides the usual setup for LSH, we add two components: (i) a scheduler deciding if an LSH update should be performed in this iteration (ii) if updating, a learning (data-adaptive) mechanism will be triggered for tuning the LSH hash functions based on the current weights, and then LSH updates will be performed.

dient steps are large enough to cause the model parameters' LSH hash codes to change. Therefore, we count the number of hash code changes that occur for important models such as Transformers and fully-connected NNs from previous work (Chen et al., 2020; Kitaev et al., 2020). We find that only 1% of the hash codes change after each epoch on average for Transformers, and 5% for a fully-connected NNs, so most weights do not change on a scale that is enough to trigger an LSH update. Furthermore, the rate of hash code change initially decays exponentially and eventually plateaus, as shown later in Figure 3. We calculate that a **100×** speed up is possible with an update oracle. In contrast, current approaches with LSH have not fully exploited this observation. We show in Section 4.2 that they either blindly skip updates for speed, resulting in a 10-point accuracy drop, or suffer from a $20\times$ slow-down.

We demonstrate that these slowly changing hash codes provide two opportunities to realize the ideal data-dependent LSH framework described above (shown as the intuition for the scheduler in Figure 1). First, given weight changes, an algorithm could adaptively schedule LSH updates to realize low update overhead. Second, given current weights, we could learn data-dependent hash functions for better LSH bucket distribution to achieve fast query time. However, this comes with three challenges: (i) how to characterize the slowly-changing phenomenon without computing hash codes, (ii) how to schedule the updates without the oracle, and (iii) how to learn data-dependent hash functions for shorter query time without compromising on the update time.

In this paper, we first provide a general formulation of the problem as *dynamic NNS* in Section 2. In Section 3, we propose MONGOOSE, a framework for fast and memory-efficient NN training to address the above challenges. We show one major observation, *slow change*, and two key components based on it in MONGOOSE. Specifically,

- In Section 3.1, we measure the $\ell_2$ norm of the weight changes and their hash code changes during training on a fully-connected NN. (More models are studied in Appendix A). We find that (i) the hash codes are slowly changing (ii) both quantities share similar trajectories, but the former's slow change is the necessary condition of the latter's. Therefore, we formally define the slow change of $\ell_2$ norm of weight changes in Assumption 3.1, which is the key to build MONGOOSE.
- In Section 3.2, we present an algorithm for scheduling efficient LSH updates. We closely analyze our scheduler's theoretical guarantees and provide an upper bound of its running time, which is proportional to the $\ell_2$ norm of the weight changes. Therefore, under the *slow change* assumption, we show that our scheduler is provably faster than previous approaches.
- In Section 3.3, we propose a method of learning parameterized LSH hash functions (e.g., SimHash (Charikar, 2002)) during NN training. Our method utilizes intermediate activations during the forward pass to generate training inputs for tuning LSH hash functions with little overhead. Combining this with our scheduler further decreases the overhead, while providing hash functions that better separate data.

Finally, in Section 4, we demonstrate the efficacy of MONGOOSE on two LSH-NN systems, SLIDE and Reformer. For SLIDE applications, we show up to $6.5\times$ speedup in time and 8% higher accuracy on three datasets. For Reformer applications, we show improvement in perplexity when training lan-

guage modeling benchmarks from scratch. Moreover, We provide two additional sets of experiments to show how MONGOOSE addresses the aforementioned three challenges separately.

## 2 RELATED WORK AND PROBLEM SETTING

In this section, we first discuss applications of LSH in the NN training setting and introduce data-dependent LSH techniques. Then we formally define the problem we solve as *dynamic NNS*.

### 2.1 LSH FOR EFFICIENT NN TRAINING

Recent works take advantage of LSH as an efficient NNS algorithm to speed up matrix multiplication in neural networks. SLIDE (Chen et al., 2020) is an algorithm that retrieves neurons with maximum inner product during the forward pass via an LSH-based data structure. In this way, in the backward pass gradients are only computed for neurons with estimated large gradients. Reformer (Kitaev et al., 2020), a variant of Transformer, similarly uses LSH to reduce the memory bottleneck of self-attention layers over long sequences. One pain point of LSH in this setting is that model weights change throughout training. This necessitates constantly updating the LSH data structure. Figure 2 illustrates what happens when we *don't* perform such updates. In short, failing to update the LSH data structure as the search data changes degrades its NNS performance. This in turn worsens the quality of the matrix product approximation. In our

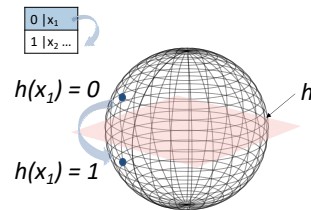

Figure 2: Visualization of why LSH updates are essential when data changes. $h$ is a hyperplane separating two hash buckets. If $x_1$ is updated in training, its hash value may change.

experiments, we found that failing to update the LSH data structure in SLIDE causes a 28% decrease in top-1 accuracy for a fully-connected NN. (More related work is presented in Appendix E.1)

### 2.2 PROBLEM FORMULATION

In this section, we formulate LSH for effcient training as a *dynamic NNS* problem. This formulation is closely built on the static NNS problem and the well-known lower bound on LSH complexity. In the static NNS problem, we are given a set of weights $w_1, \cdots, w_n \in \mathbb{R}^d$ and want to construct a data structure (NNS-ds) that supports the QUERY operation, defined as follows: given $x \in \mathbb{R}^d$, QUERY$(x)$ returns a set $S$ of weights $w_i$ that are all "close" to $x$ in a distance measure.

More precisely, we require the QUERY$(x)$ operation to be $(c_1, c_2)$-accurate:

**Definition 2.1** ($(c_1, c_2)$-accurate). *Denote $S$ to be the (random) set returned by* QUERY$(x)$. *We say* QUERY$(x)$ *is* $(c_1, c_2)$ *accurate if*

*(i) for any $i \in [n]$ such that $\langle w_i, x \rangle \geq c_1 \|x\|_2 \|w_i\|_2$, $\Pr[w_i \in S] \geq 1 - 1/\operatorname{poly}(n)$,*
*(ii) for any $i \in [n]$ such that $\langle w_i, x \rangle < c_2 \|x\|_2 \|w_i\|_2$, $\Pr[w_i \in S] < 1/\operatorname{poly}(n)$.*

In our application, we assume $1 > c_1 > c_2 > 0$ to be some constants and denote $\rho = (c_2/c_1)^2$. In the dynamic NNS problem, the weights $w_1, \cdots, w_n \in \mathbb{R}^d$ can evolve over time, so we need to update the data structure.

**Lemma 2.2** ((Andoni & Indyk, 2006)). *Using LSH, one can achieve $(c_1, c_2)$ accuracy with query time $O(dn^\rho)$, preprocessing time $O(dn^{1+\rho})$, updating time $O(dn^\rho)$.*

## 3 MONGOOSE: A FRAMEWORK FOR LEARNABLE LSH

We present the workflow of our main framework in Figure 1. In Section 3.1, we first show the key observation that encourages the design of MONGOOSE and formally define the *slow change* assumption. In Section 3.2, we introduce our first component, an algorithm for scheduling LSH updates with provable guarantees based on the assumption. Finally, we present the second component, low-cost learnable LSH hash functions, in Section 3.3. More details about the efficient implementation of MONGOOSE are in Appendix F.

### 3.1 CHARACTERIZATION OF "SLOW CHANGE" PHENOMENON

We first present an observation on model weights and how their LSH hash codes change during training. Based on this observation, we define the *slow change* assumption. We denote $\Delta W$ as the

difference between weights across gradient steps, and $\Delta H$ as the Hamming distance between hash codes during training.

**Observation.** We plot the $\ell_2$ norm of $\Delta W$ and $\Delta H$ of the weights during training a one layered fully-connected NN in Figure 3. Our most surprising finding is that only 5% of hash codes change after each epoch on average, which implies that for most neurons, $\Delta W$ is not large enough to trigger hash code changes. For both $\Delta W$ and $\Delta H$, there is a sharp drop at the early stages of training before they plateau. (Similar observations on Transformers and further discussions are presented in Appendix A.)

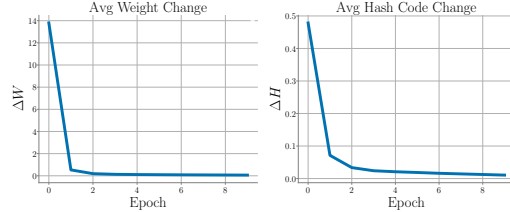

Figure 3: $\Delta W$ and $\Delta H$ both start out relatively high at the beginning of training, but quickly drop off and flatten out.

**Insights.** In the *dynamic* NNS problem, input data (model weights) change over time. Without any assumptions about weight updates, naïvely applying LSH will require updating the LSH hash functions at every time step. However, the above observation suggests that we can reasonably assume $\Delta W$ is (roughly) upper-bounded by $\Delta H$. Denote $w$ as the weight matrix, $n$ as number of neurons and $w_i$ as the weight vector for neuron $i$. Formally:

**Assumption 3.1** (Slow change). *Assume NN weights change slowly over time. In particular, we assume there is an upper bound on the expected movement of the weights of the neural networks ($C_1$) and an upper bound on the variance ($C_2$). Specifically, we denote the initial weight matrix as $w^{(0)} \in \mathbb{R}^{n \times d}$. We assume the (random) update sequence $w^{(1)}, \cdots, w^{(T)} \in \mathbb{R}^{n \times d}$ satisfies*

$$\sum_{i=1}^{n} \left\| \mathbb{E}[w_i^{(k+1)}] - w_i^{(k)} \right\|_2^2 \leq C_1^2 \qquad and \qquad \sum_{i=1}^{n} \|\text{Var}[w_i^{(k+1)}]\|^2 \leq C_2^2 , \tag{1}$$

*where the expectation and variance is conditioned on $w_i^{(k)}$ for all $k = 0, 1, \cdots, T - 1$.*

### 3.2 A SMART SCHEDULER FOR LEARNABLE LSH UPDATES

In this section, we introduce a dynamic data structure to schedule updates for LSH. Our scheduler provides significant speedup under two realistic assumptions without compromising the theoretical guarantee of LSH: (1) *slow change* defined in Assumption 3.1, (2) *batch speed up* defined in Assumption 3.2. The scheduler can generalize to any near-neighbor search data structure (NNS-ds), but we show a design for LSH as an example since that is the NNS-ds in MONGOOSE.

#### 3.2.1 SMART SCHEDULER

The pseudo-code for our smart scheduler is shown in Algorithm 1. Our scheduler is inspired by a dynamic data structure proposed in a recent line of work (Cohen et al., 2019), which was originally designed for inverse maintainance in Linear Programming solvers. We provide a novel application here for the dynamic LSH problem.

**Notation:** We denote $[n] = \{1, \cdots, n\}$. The $\widetilde{O}(\cdot)$ notation hides polylogarithmic

---

**Algorithm 1** General Maintenance Data Structure

1: **Input:** $w \in \mathbb{R}^{n \times d}$, $v \in \mathbb{R}^{n \times d}$, $\epsilon_{\text{mds}} \in (0, 1/\log^2 n)$, $a = \min\{\rho, \alpha\}$, $S \subseteq [n]$
2: **procedure** INITIALIZE($w, \epsilon_{\text{mds}}$)                    ▷ Lemma B.2
3:     $w \leftarrow w, v \leftarrow w, \epsilon_{\text{mds}} \leftarrow \epsilon_{\text{mds}}, S \leftarrow \emptyset$,
4:     Build LSH with accuracy $(c_1 - \epsilon_{\text{mds}}, c_2 + \epsilon_{\text{mds}})$
5: **end procedure**
6: **procedure** UPDATE($w^{\text{new}}$)                    ▷ Lemma B.3
7:     $y_i \leftarrow w_i^{\text{new}} - v_i, \forall i \in [n]$
8:     $r \leftarrow$ the number of indices $i$ that $\|y_i\|_2 \geq \epsilon_{\text{mds}}/2$.
9:     **if** $r < n^a$ **then**
10:         $v^{\text{new}} \leftarrow v$
11:     **else**
12:         Let $\pi : [n] \rightarrow [n]$, that $\|y_{\pi(i)}\|_2 \geq \|y_{\pi(i+1)}\|_2$
13:         **while** $1.5 \cdot r < n$ and $\|y_{\pi(\lceil 1.5 \cdot r \rceil)}\|_2 \geq (1 - 1/\log n)\|y_{\pi(r)}\|_2$ **do**
14:             $r \leftarrow \min(\lceil 1.5 \cdot r \rceil, n)$
15:         **end while**                    ▷ Finding a smooth cut
16:         $v_{\pi(i)}^{\text{new}} \leftarrow \begin{cases} w_{\pi(i)}^{\text{new}} & i \in \{1, 2, \cdots, r\} \\ v_{\pi(i)} & i \in \{r+1, \cdots, n\} \end{cases}$
17:         LSH.UPDATE $\pi(1), \cdots, \pi(r)$
18:     **end if**
19:     $w \leftarrow w^{\text{new}}, v \leftarrow v^{\text{new}}$,
20:     $S = \{i \in [n] : \|w_i - v_i\|_2 \geq \epsilon_{\text{mds}}/2\}$
21: **end procedure**
22: **procedure** QUERY($h$)                    ▷ Lemma B.4
23:     LSH.QUERY and check $S$
24: **end procedure**

---

dependence on $n, d, \epsilon_{\mathrm{mds}}$. We use $w \in \mathbb{R}^{n \times d}$ to denote the weights of the neural network and each row $w_i \in \mathbb{R}^d$ ($i \in [n]$) corresponds to the input weight of the $i$-th neuron. We maintain a copy of the neural network weights, denoted as $v \in \mathbb{R}^{n \times d}$, that approximately equals $w$ except for a few coordinates. We take $\epsilon_{\mathrm{mds}}$ to be a sufficiently small precision parameter, i.e. $\epsilon_{\mathrm{mds}} \ll c_1 - c_2$, which satisfies $\epsilon_{\mathrm{mds}} \in \Omega(1/\log^2 n)$. We use $S$ to denote the set of indices where $w$ and $v$ differ significantly (see Line 20). LSH.UPDATE (see line 17) takes indices of the neural weights and updates the LSH hashcode correspondingly.

**Overview of algorithm:**   During the training process, we maintain two copies of the NN weights $w, v$, where $w$ are the actual weights that are updated via gradient descent and $v$ approximately equals $w$ except for a few coordinates. We set an empirical threshold of $\epsilon_{\mathrm{mds}}$. Due to the robustness of the NNS problem, we can tolerate this small error $\epsilon_{\mathrm{mds}}$ such that only those coordinates with a difference greater than $\epsilon_{\mathrm{mds}}$ matter. To exploit the benefit of batch updating, we perform lazy updates on the LSH hash table. In particular, we update the LSH hash table if and only if the difference between $w, v$ exceeds some threshold number $n^\rho$ (see line 9 to line 11). One key part of our update procedure is that we also update some low-error coordinates when there are *too many* of them (see line 13 to line 16). This procedure can be seen as forecasting future updates. The intuition is that when there are too many coordinates close to the decision boundary, they are likely to change within a short amount of time. Updating these all together could further take advantage of the batch updates. To answer an LSH query, we keep track of a short list $S$ (see line 20), which records coordinates that differ significantly between $w, v$. We can interpret $S$ as the indices of corrupted datapoints to keep track of when performing an LSH query, since they are no longer the same. In the query procedure (see line 22 to line 24), besides performing the normal LSH query, we also check $S$ and pick up weights that are close to $x$. Hence, we balance the overhead of query time with update time, where update time is usually the bottleneck.

### 3.2.2   THEORETICAL ANALYSIS

**Assumption 3.2** (batch speedup). *Denote $T_r$ as the updating time for LSH when the weights of $r$ neurons change and $T_r = r \cdot t_r$. That is, $T_r$ is the updating time when we update $r$ entries of the LSH hash table and $t_r$ is the average updating time. We assume $\{t_i\}_{i=1,\dots n}$ is non increasing, i.e., $t_1 \geq t_2 \geq \dots \geq t_n$. We further assume $T_r = n^\rho$ when $r \leq n^\alpha$ for some constant $\alpha$.*

The batch speedup assumption is realistic, especially due to advances in support for massive parallelism on modern computation architectures.

**Theorem 3.3** (LSH maintenance). *For any constant $c_1, c_2$ ($c_1 > c_2$), there is a dynamic data structure (Algorithm 1) for LSH maintenance that achieves $(c_1, c_2)$-accuracy. The data structure takes $\widetilde{O}(dn^{1+\rho})$ time to initialize and each call of QUERY($h$) takes time $\widetilde{O}(n^\rho d)$. By taking $a = \min\{\rho, \alpha\}$ and*

$$g_r = \begin{cases} n^{\rho-a}, & r \leq n^a; \\ t_r, & r > n^a. \end{cases}$$

*The amortized expected time per call of UPDATE($w$) is at most*

$$\widetilde{O}((C_1 + C_2) \cdot \|g\|_2). \tag{2}$$

**Remark 3.4.** *Depending on the exact form of batch updating time $t_r$, the (amortized) running time and the benefit of our scheduler could be different. We provide concrete examples in Appendix B.7, showing advantages of our smart scheduler under different scenarios. In general, we remark that (1) Our scheduler is always better than the naïve sequential updating strategy. (2) Our scheduler is* oblivious *to the exact form of the update time $t_r$ and the weight change $C_1, C_2$. That is to say, we do not need to fine tune these parameters. In NN training, the only parameter that requires fine tuning is the precision parameter $\epsilon_{\mathrm{mds}}$.*

### 3.3   HASH FUNCTION LEARNING

In this section, we provide a simple streaming algorithm to learn parameterized hash functions. Here, the learning objective is to track the weight distribution as the weights change to better separate the

input data to the NNS problem. Recall that in last section, we show an algorithm for scheduling the LSH updates, which makes it possible to involve learning for LSH and improve the query efficiency. Therefore, the overall update overhead can be reduced even with learning overhead if scheduled well, since the update time is closely related to query time.

### 3.3.1 LEARNABLE LSH

We view the sequence of neural network weights throughout training as a matrix stream. In each training iteration, if our scheduler schedules an update, we compute a training signal with which we can train our LSH parameters. Specifically, we compute the classical triplet loss (Chechik et al., 2010) using positive and negative samples from the current model weights for each LSH query. Here, positive samples (denoted by $\mathcal{P}_+$) are neurons that have higher inner product within the input retrieved by LSH, while negative samples (denoted by $\mathcal{P}_-$) are neurons with lower inner product that are selected. Denote a set of neurons in a particular layer as $\mathcal{C} = \{v_r \mid 0 \leq r < m\}$ and the input embedding as $q_i$. Denote the set of

---

**Algorithm 2** Learnable Hash Function

1: **Input: query set** $Q = \{q_i\}_{i \in [N]}$, **query size** $N$
2: **hash table** $\mathcal{G} = \{g_l\}_{l \in [L]}$
3: **hash function** $\forall l \in [L]$, $\mathcal{H}_l = \{h_{l,k}\}_{k \in [K]}$, $\mathcal{H} = \cup_{l \in [L]} \mathcal{H}_l$
4: **Retrieved Neurons Sets** $\{\mathcal{S}_i\}_{i \in [N]}, \mathcal{S}_i = \{v_r\}_{r \in [M]}$
5: **parameters** $t_+$, $t_-$
6: $\mathcal{P}_+ \leftarrow \emptyset, \mathcal{P}_- \leftarrow \emptyset$ $\qquad \triangleright \mathcal{P}_+, \mathcal{P}_- \in \mathbb{R}^{2d+2}$
7: **for** $i = 1 \rightarrow N$ **do**
8: $\qquad \mathcal{P}_+ \leftarrow \mathcal{P}_+ \cup \{(q_i, v_r) \mid v_r \in S_i, \langle q_i, v_r \rangle > t_+\}$
9: $\qquad \mathcal{P}_- \leftarrow \mathcal{P}_- \cup \{(q_i, v_r) \mid v_r \in S_i, \langle q_i, v_r \rangle < t_-\}$
10: **end for**
11: $\text{size}^{\text{new}} \leftarrow \min\{|\mathcal{P}_+|, |\mathcal{P}_-|\}$
12: $\text{RESIZE}(\mathcal{P}_+, \mathcal{P}_-, \text{size}^{\text{new}})$
13: $\mathcal{H}^{\text{new}} \leftarrow \text{REHASH}(\mathcal{P}_+, \mathcal{P}_-, \mathcal{L}) \triangleright \mathcal{L}$ is a loss function
14: $\mathcal{G}^{\text{new}} \leftarrow \text{REBUILD}(\mathcal{G}, \mathcal{H}^{\text{new}})$
15: **return** $\mathcal{G}^{\text{new}}, \mathcal{H}^{\text{new}}$

---

neurons retrieved from the LSH hash tables using $q_i$ as $\mathcal{S}_i$. We formally present the learning algorithm in Algorithm 2. The operation of updating or training LSH hash functions is defined as REHASH and the operation of updating the hash tables based on the updated hash functions and weights is defined as REBUILD. The triplet loss is defined as follows:

$$\mathcal{L}(\mathcal{H}, \mathcal{P}_+, \mathcal{P}_-) = \max(0, \sum_{(q,v) \in \mathcal{P}_+} - \cos(\mathcal{H}(q), \mathcal{H}(v)) + \sum_{(q,v) \in \mathcal{P}_-} \cos(\mathcal{H}(q), \mathcal{H}(v)) + \alpha), \quad (3)$$

where $\mathcal{H}$ denotes the LSH hash function at the current iteration and $\alpha$ is a margin between positive and negative pairs. $\cos(\mathcal{H}(\cdot), \mathcal{H}(\cdot))$ represents the cosine similarity. Note that in the application setting of unit norm like (Kitaev et al., 2020), using inner product is equivalent to cosine similarity. We provide a detailed justification of this learning approach in Appendix C.1.

### 3.3.2 FURTHER SPEEDUP

We demonstrate how to optimize learnable LSH for further acceleration in NN training. Here, we provide the main insights; further details are in Appendix A.

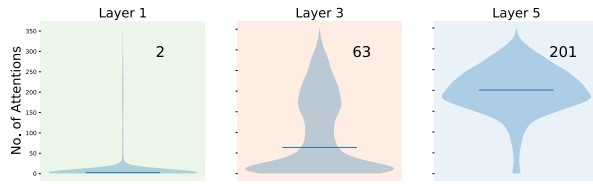

Figure 4: Attention distribution examples in different layers: Y-axis is the least number of attentions summing to 90% of full attentions. The annotations represent the median. The total number of attentions (or sequence length) is 512.

**Layer Selection:** In previous LSH-NN methods such as SLIDE or Reformer, the designation of whether a layer uses LSH is treated as a tunable hyperparameter. For example, Reformer applies LSH in three out of twelve attention layers in their final model based on empirical performance. In MONGOOSE we make this designation in a more principled manner. The intuition behind the design can be presented by visualizing the statistics of attention weights in different layers of the Transformer model (Figure 4). The three plots represent the distribution of the average number of coordinates with "heavy" attention for each layer. We also calculate the size of the minimal set of attention weights that captures 90% of the total attention ((Ramsauer et al., 2020) perform a similar experiment). The median for the left plot is 2, meaning that they are well-separated and LSH can perform well on such a distribution. On the other hand, learnable LSH can be very helpful for the right plot, because classical LSH is inefficient for this distribution. Therefore, we can use this observation when deciding between LSH or learnable LSH in each layer for later training stages or fine-tuning on downstream tasks.

Table 1: This table summarizes the performance of MONGOOSE, SLIDE and Full-Computation implemented with PyTorch (Paszke et al., 2019). $P@1/5$ is top-1/5 precision. Time represents convergence time and Mem represents memory consumption.

| Datasets | Full-Computation | | | | SLIDE | | | | MONGOOSE | | | |
|---|---|---|---|---|---|---|---|---|---|---|---|---|
| | $P@1$ | $P@5$ | Time (s) | Mem (GB) | $P@1$ | $P@5$ | Time | Mem | $P@1$ | $P@5$ | Time | Mem |
| Wiki10-31K | **0.824** | 0.578 | 63 | 0.3 | 0.824 | 0.556 | 47 (1.3 ×) | 0.24 (1.3×) | **0.824** | **0.618** | **35 (1.8×)** | **0.2 (1.5×)** |
| Delicious-200K | 0.446 | 0.356 | 483 | 2.2 | 0.465 | **0.372** | 318 (1.5×) | 1.7 (1.3×) | **0.469** | 0.371 | **162 (3×)** | **1.5 (1.5×)** |
| Wiki-325K | 0.501 | 0.235 | 5702 | 3.9 | 0.438 | 0.205 | 4680 (1.2×) | 3.3 (1.2×) | **0.519** | **0.255** | **1906 (3×)** | **2.7 (1.5×)** |
| Amz-670K | 0.332 | 0.273 | 8310 | 6.7 | 0.335 | 0.276 | 3224 (2.6×) | 4.3 (1.6×) | **0.336** | **0.281** | **1279 (6.5×)** | **3.3 (2×)** |

From a system design perspective, MONGOOSE improves the flexibility of LSH's usage in NN training. We provide a smart scheduler to decide when to perform LSH updates and visualization on when to use LSH or learnable LSH. Therefore, the training process has the following tunable parameters for LSH: (i) the layers which we apply random LSH to, (ii) the layers for which we can use learning to improve LSH's random hash functions, (iii) the runtime that LSH or learnable LSH should perform the update in. Setting these parameters appropriately will enable much more efficient NN training or fine-tuning, as supported by our theory and observations.

## 4 EXPERIMENTS

In Section 4.1 we present the results of experiments that demonstrate the higher performance of MONGOOSE over other LSH-NN frameworks during training. Then we study the effectiveness of the two components: (i) smart scheduler in Section 4.2, (ii) learnable LSH in Section 4.3 [1].

### 4.1 MAIN RESULTS: MONGOOSE IN NN TRAINING

We present the results of main experiments that demonstrate MONGOOSE's advantage in end-to-end NN training over two different LSH-NN baselines on recommendation and language modeling tasks.

### 4.1.1 TASK1: EXTREME MULTI-LABEL CLASSIFICATION

**Settings:** We compare the performance of MONGOOSE against SLIDE and networks without LSH (Full-Computation) on 3 large-scale multi-label classification datasets: Wiki10-31k, Delicious-200k, Wiki-325k and Amz-670k. The datasets are obtained from the Extreme Classification Repository (Bhatia et al., 2016), which is a benchmark for various recommendation systems. The model is a fully-connected network with one hidden-layer. The statistics of each dataset are in Appendix D. We report the best results of each framework in $P@1/5$ following metrics in (Bhatia et al., 2016).

**Results:** Table 1 shows the test accuracy, wall-clock time to converge and memory consumption for MONGOOSE and other baselines. Left plot in Figure 5 visualizes $P@1$ for MONGOOSE and other baselines during the training process. MONGOOSE achieves 1.8× faster convergence and 1.5× less memory consumption compared to Full-Computation on Wiki10-31K, 3× faster convergence, 1.5× less memory on Delicious-200K and Wiki-325K and 6.5× faster convergence, 2× less memory on Amz-

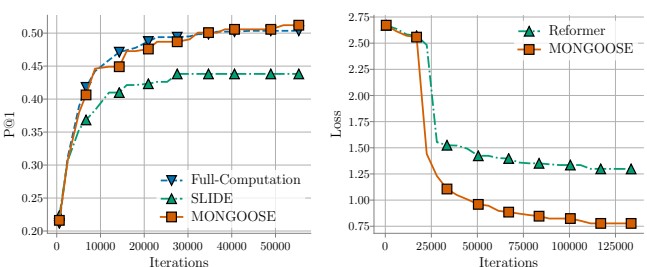

Figure 5: Left plot: $P@1$ versus training iteration on Wiki-325k. Right plot: Loss versus training iteration on a synthetic copying task.

670K. SLIDE has a large accuracy drop without a substantial speedup for Wiki-325k, but our method has a 3× speedup with no accuracy loss.

**Analysis:** The above mainly focuses on MONGOOSE's end-to-end performance during training. We also present closer analysis of a single linear layer during training in Table 2. In isolation, we find that MONGOOSE achieves up to 20× faster convergence and 4.5× less memory consumption compared to Full-Computation. The greater

Table 2: Time and memory advantage compared to Full-Computation on one layer.

| Datasets | SLIDE | | MONGOOSE | |
|---|---|---|---|---|
| | Time | Mem | Time | Mem |
| Wiki10-31K | 1.8× | 2× | **6.6×** | **4×** |
| Delicious-200K | 1.8× | 2.2× | **8.6×** | **4.5×** |
| Wiki-325K | 1.4× | 1.7× | **20 ×** | **4×** |

---

[1]MONGOOSE code is available at https://github.com/HazyResearch/mongoose

time and memory savings here compared to Table 1 are because we ignore the earlier large embedding layer of the model (Details are in Appendix D.3). We make three observations to help explain the memory and time savings from MONGOOSE. First, matrix multiplication is only performed on a small subset of neurons. The average number of neurons sampled in the output layer for MONGOOSE on Delicious-200K is around 6k compared to the total number of 200k. Second, our scheduler further reduces updating costs. On Wiki-325K, we observe updating frequency drops significantly at the start of the second epoch. This is compatible with our observation of slow weight change in Figure 3. Finally, learnable LSH improves LSH query efficiency and quality. In addition, we observe that MONGOOSE obtains a slightly higher accuracy compared to Full-Computation, which is a side benefit from activation sparsity (Ba & Frey, 2013).

### 4.1.2  TASK2: EFFICIENT TRANSFORMER

**Settings:** We run experiments on two tasks from Reformer (Kitaev et al., 2020). The first task is a synthetic sequence duplication task where inputs are of the form $0w0w$ and $w \in \{0, ..., N\}^*$. This task is useful for demonstrating the effectiveness of LSH attention: it requires non-local attention lookups and therefore cannot be solved by any model relying on sparse attention with a limited range (*e.g.*, local attention). We compare MONGOOSE with a 1-layer Reformer by replacing the data-independent LSH in the attention layer with learnable LSH, updating the LSH parameters according to our smart scheduler. The second task is a standard language modeling task on the enwik8 dataset. For this second task, we use sequence length of 8192 and 10 LSH-attention layers.

**Results:** We compare the test loss of MONGOOSE and Reformer on each task. The loss curve for the synthetic sequence duplication task is shown in Figure 5 (right). On this task, MONGOOSE achieves a loss value of 0.75, which is 0.5 better than Reformer. Moreover, MONGOOSE only needs 20% time to reach the same loss as Reformer ($5\times$ speed-up). Similarly, on enwik8, Reformer only reaches 2.65 in perplexity, while MONGOOSE achieves 2.59 in the same setting. Both MONGOOSE and Reformer (with LSH) save $6\times$ memory over the original Transformer in above two tasks (more details are in Appendix D.4).

### 4.2  STUDY ON SMART SCHEDULER

In this section, we demonstrate the superiority of our smart scheduling algorithm over the two baselines we mentioned in Section 1 on Wiki-325K.

Table 3: Speed with respect to accuracy

|  | $P$@1 | $P$@5 | Speed (batch/s) |
|---|---|---|---|
| Infrequent Scheduler | 0.39 | 0.193 | 31.2 |
| W/O Scheduler | 0.521 | 0.254 | 0.4 |
| MONGOOSE Scheduler | 0.519 | 0.255 | 7.7 |

**Effect of Smart Scheduler:** The first approach ("Infrequent Scheduler") performs an update on a fixed schedule, every 1000 iterations (6 times per epoch on Wiki-325K). The other baseline performs an update every iteration ("W/O Scheduler"). We compare speed (number of batches processed per second) and accuracy ($P$@1 and $P$@5) of these schedulers in Table 3. On average, MONGOOSE Scheduler reduces more than **97**% of the update overhead compared to W/O Scheduler, leading an end-to-end $20\times$ speed-up. Meanwhile, although Infrequent Scheduler is faster in speed, it fails to reach the same task accuracy as the other approaches. In fact, MONGOOSE Scheduler achieves Infrequent Scheduler's final accuracy in **66**% less time. We can see that MONGOOSE substantially reduces LSH update overhead while maintaining performance.

### 4.3  STUDY ON LEARNABLE LSH

**Effect of Learnable LSH:** In this section, we compare the NNS performance of MONGOOSE's learnable LSH scheme to classical LSH on their NNS ability using the same number of hash functions. In Figure 6 (left) we plot the difference between average inner products computed using neurons retrieved by learnable LSH (MONGOOSE) and random LSH (SLIDE) and inner products using the same number of randomly chosen neurons. A negative value on these plots indicates that LSH is performing worse than random sampling. We can see that at the beginning of training, when weights are initialized randomly, data-independent LSH is sufficient to partition the data. However, as training progresses, SimHash used in SLIDE fails to adapt to the evolving weight distribution, while our selection better approximates the full softmax.

We conduct the same experiments for enwik8 language modeling. In Figure 6 (right), we can see that data-dependent hashing computes higher inner products within each bucket as training progresses.

Table 4: This table compares MONGOOSE and HNSW. $P@1/5$ is the top-1/5 precision.

| | Rebuild Time(s) vs Number of Neurons | | | Wiki-10k Training | | | Wiki-325k Training | | |
|---|---|---|---|---|---|---|---|---|---|
| | 10k | 100k | 300k | $P@1$ | $P@5$ | Time (s) | $P@1$ | $P@5$ | Time (s) |
| HNSW | 0.4983 | 5.1168 | 20.5933 | **0.829** | 0.602 | 60 | 0.4625 | 0.2052 | 10106 |
| MONGOOSE | **0.4625 (1.1×)** | **0.4780 (11×)** | **0.8610 (23×)** | 0.824 | **0.618** | **35 (1.7×)** | **0.519** | **0.255** | **1906 (5.3×)** |

For attention layers in the Transformer models, random LSH fails to approximate attention well in certain layers due to a skewed attention distribution (this phenomenon was also explained in Section 3.3.2 and Figure 4). In attention layers, our selection produces a better characterization of the full attention.

**Other data-dependent NNS data structures:** In Section 1, we have briefly discussed that most existing data-dependent NNS-ds have high update costs. We choose an LSH-based data structure in MONGOOSE because update overhead for hash-table-based NNS-ds is less compared to other methods (e.g. graph-based) in practice (Wang et al., 2018; Gong et al., 2020). For example, in a recent paper, (Dong et al., 2019)

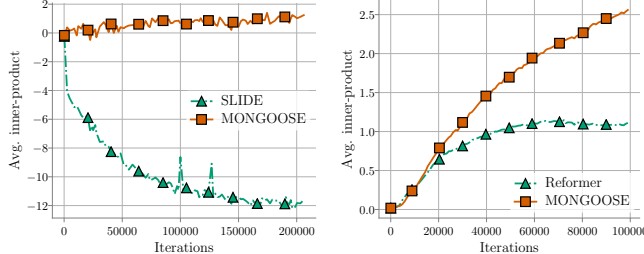

Figure 6: The normalized inner product computed by NNS-ds neurons. In both Wiki-325k (left) and enwik8 (right), MONGOOSE has much higher average inner product than baselines.

proposed a $k$-nearest-neighbor graph-based algorithm for NNS that requires several hours to process 1 million data points. Although a recent work, HNSW (Malkov & Yashunin, 2018), is relatively fast in preprocessing and updating step among graph-based data-dependent NNS-ds empirically, it does not have guarantees (Fu et al., 2017). In Table 4 we compare the update times of the learnable LSH and HNSW, and their performance during training when both are equipped with the MONGOOSE smart scheduler. We can see that HNSW's updating overhead grows exponentially and becomes over 20× learnable LSH's overhead in 350k neurons layer. However, it is an ongoing work to reduce the updating overhead of graph-based NNS-ds, so it could be potentially included in MONGOOSE later.

## 5 CONCLUSION

In this work, we make a novel observation on the slowly changing LSH hash codes during LSH-NN training. Based on the observation, we present MONGOOSE, a learnable LSH framework that outperforms original LSH methods in efficient neural network training. We demonstrate two key designs in the framework. One is a smart scheduling algorithm which reduces the LSH update overhead. The other one is a low cost learnable LSH which can improve the query efficiency and hashing quality. We empirically verify the effectiveness of our framework on both recommendation and language modeling tasks. The slowly changing observation along with the smart scheduler in MONGOOSE opens up the opportunity of designing new NNS-ds for dynamic similarity search for efficient NN training.

ACKNOWLEDGMENTS

We would like to thank Xun Huang, Fred Sala, Avner May, Ben Coleman and the anonymous reviewers for helpful discussions and feedback. We gratefully acknowledge the support of NIH under No. U54EB020405 (Mobilize), NSF under Nos. CCF1763315 (Beyond Sparsity), CCF1563078 (Volume to Velocity), and 1937301 (RTML); ONR under No. N000141712266 (Unifying Weak Supervision); the Moore Foundation, NXP, Xilinx, LETI-CEA, Intel, IBM, Microsoft, NEC, Toshiba, TSMC, ARM, Hitachi, BASF, Accenture, Ericsson, Qualcomm, Analog Devices, the Okawa Foundation, American Family Insurance, Google Cloud, Swiss Re, Total, the HAI-AWS Cloud Credits for Research program, the Stanford Data Science Initiative (SDSI), and members of the Stanford DAWN project: Facebook, Google, and VMWare. The Mobilize Center is a Biomedical Technology Resource Center, funded by the NIH National Institute of Biomedical Imaging and Bioengineering through Grant P41EB027060. The U.S. Government is authorized to reproduce and distribute reprints for Governmental purposes notwithstanding any copyright notation thereon. Any opinions, findings, and conclusions or recommendations expressed in this material are those of the authors and do not necessarily reflect the views, policies, or endorsements, either expressed or implied, of NIH, ONR, or the U.S. Government.

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

# Appendix

## Table of Contents

## A    SLOW CHANGE OBSERVATIONS

In this section, we analyze the relative distance between iterates of model parameters and connect it with LSH updates throughout training. We target at making full use of our slowly change characterization for MONGOOSE.

**Settings.**  We report $\Delta W$ throughout training on several benchmark models.  We follow the standard training procedures of fully-connected networks on Wiki325k (Partalas et al., 2015) and Transformer (Vaswani et al., 2017) on enwik8. We choose $W$ in the last linear layer for the fully-connected model and $W$ in the projection layer before attention for the transformer model to correspond experiments in section 4.

**Results.**  We plot our results in Figure 3.  The left-two and right-most plots show that during the initial steps of the training, $\Delta W$ is relatively high, but quickly drops and flattens out afterwards. The middle and right-most plots exhibit the hash code change of $W$ in the hamming distance along with the training. The pattern matches with $\Delta W$ but has a direct connection with the LSH update overhead.  This is also consistent with LSH theory that the hash code collision probability of two vectors equals to the vectors' certain similarity, *e.g.*, angular. Note the above observations are made based on angular distance LSH. The above phenomenon suggests that if there exists an optimal scheduler to update the data structures adaptively based on the actual demand, the overhead by LSH updates can be largely reduced. Furthermore, this opens the door to make LSH learnable in order to improve query efficiency.  Since the LSH update time is also closely related to the query time, the overall update overhead might still be reduced after considering learning costs, if the updates are well scheduled. Several related works (Jastrzebski et al., 2018; Agarwal et al., 2020) have also observed the phenomenon of a sharp drop of weight changes at the early stages of training. One (Jastrzebski et al., 2018) conjectures that initially SGD visits increasingly sharp regions, reaching a maximum sharpness determined by both the learning rate and the batch-size of SGD and infer that it optimizes faster while finding a good sharp region. It is also discussed in a recent paper (Agarwal et al., 2020), which connects the phenomenon to critical learning periods defined in (Achille et al., 2017). This phenomenon can be observed on text (Transformers), recommendation (Embedding Models) and image (CNN) data and thereby it is relatively general.

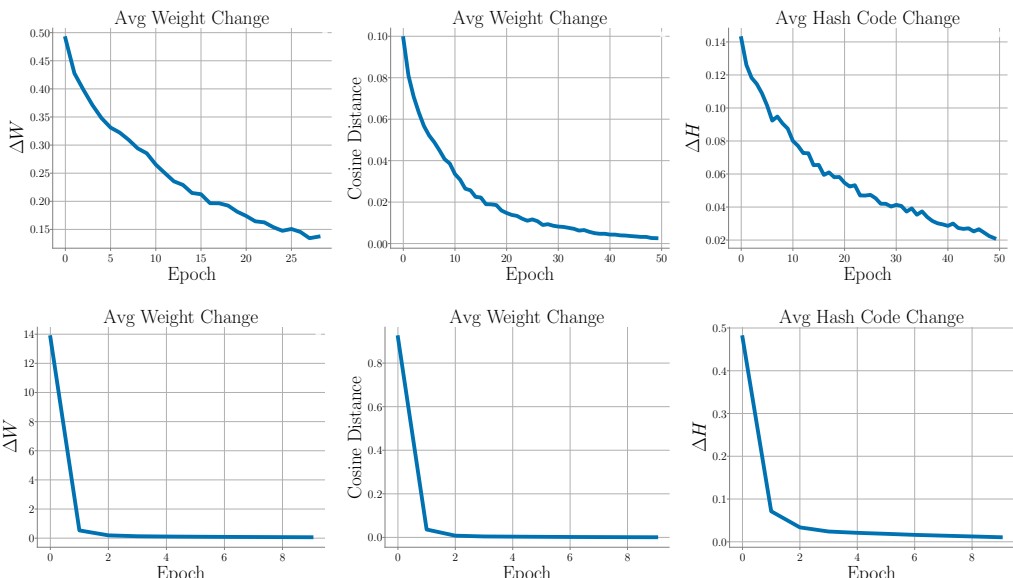

Figure 7: We show the average change of weight (left), cosine similarity (middle), weight's hash code (right). Top row is reformer and the bottom row is for Wiki-325k.

## B  DYNAMIC MAINTENANCE DATA STRUCTURE

The idea of dynamic maintenance has been successfully applied to matrix inversion problem, which served as a central component in linear program solvers (Cohen et al., 2019; Lee et al., 2019; Brand et al., 2020b; Song & Yu, 2021; Jiang et al., 2021), cutting plane method (Jiang et al., 2020b), maximum matching Brand et al. (2020a), max-flow Brand et al. (2021), and semi-definite program solvers (Jiang et al., 2020a; Huang et al., 2021). In this work, we adopt it for dynamic LSH maintenance problem. Since the underlying problem is completely different, it requires several non-trivial generalizations and more fine-grained analysis (see Lemma B.2, Lemma B.3, Lemma B.4). Moreover, in our proof, we care about the absolute distance of between vectors (i.e. $\|w_i - v_i\|_2$) rather than the relative difference between real numbers Cohen et al. (2019), and therefore, we need a new construction of the potential function (see Lemma B.8).

The rest of this section is organized as follows

- In Section B.1, we provide an illustration of the smart scheduler.
- In Section B.2, we state our main results.
- In Section B.3, we give a high-level overview on the correctness and the running time of the algorithm.
- In Section B.4, we give an analyze on the movement of $w$ vector.
- In Section B.5 we give analyze on the movement of $v$ vector.
- In Section B.6, we discussed over the choice of potential function and analysis its property.
- In Section B.7, we compare the performance our algorithm with sequential updating strategy/batch updating strategy.

### B.1  ILLUSTRATION

We present a demonstration of our smart scheduler at Figure 8. As shown in the top, if the change of data vectors (denoted as points with colors) stays within the partitions given by hash functions, the scheduler would not update current LSH tables. However, if the data vectors change such that they cross partition boundaries, the scheduler updates the hash functions.

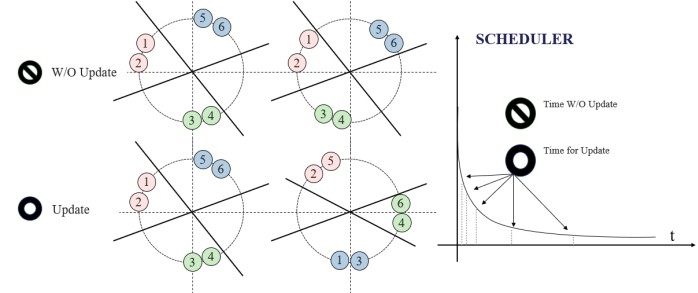

Figure 8: Illustration of the smart scheduler

### B.2  MAIN RESULT

**Theorem B.1** (Locality sensitive hashing maintenance, Formal statement of Theorem 3.3). *For any constant $c_1, c_2$ ($c_1 > c_2$) there is a dynamic data structure (Algorithm 1) that achieves $(c_1, c_2)$-accuracy. The data structure takes $\widetilde{O}(dn^{1+\rho})$ time to initialize and each call of* QUERY$(h)$ *takes time $\widetilde{O}(n^\rho d)$. By taking $a = \min\{\rho, \alpha\}$ and*

$$g_r = \begin{cases} n^{\rho-a}, & r \le n^a; \\ t_r, & r > n^a. \end{cases}$$

*The amortized expected time per call of* UPDATE$(w)$ *is at most*

$$\widetilde{O}((C_1 + C_2) \cdot \|g\|_2). \tag{4}$$

### B.3  PROOF OF THEOREM B.1

We first give a rough explanation on the proof of Theorem 3.3. For intuition, we consider the case $C_1 = \Theta(1)$, $C_2 = \Theta(1)$, and $\epsilon_{\mathrm{mp}} = \Theta(1)$ in this explanation. The amortized time analysis is based

on a potential function that measures the distance of the approximate vector $v$ and the target vector $w$. We will show that

- The cost to update the LSH data structure is proportional to the decrease of the potential.
- Each call to query increase the potential by a fixed amount.

Combining both together gives the amortized running time bound of our data structure.

Now, we explain the definition of the potential. Consider the $k$-th round of the algorithm. For all $i \in [n]$, we define $x_i^{(k)} = w_i^{(k)} - v_i^{(k)}$. Note that $\|x_i^{(k)}\|_2$ measures the distance between $w_i^{(k)}$ and $v_i^{(k)}$. Our algorithm fixes the indices with largest error $\|x_i^{(k)}\|_2$. To capture the fact that updating in a larger batch is more efficient, we define the potential as a weighted combination of the error where we put more weight to higher $x_i^{(k)}$. Formally, we sort the coordinates of $x^{(k)}$ such that $\|x_i^{(k)}\|_2 \geq \|x_{i+1}^{(k)}\|_2$ and define the potential by

$$\Psi_k = \sum_{i=1}^n g_i \cdot \psi(x_i^{(k)}),$$

where $g_i$ are positive decreasing numbers to be chosen and $\psi$ is a symmetric ($\psi(x) = \psi(-x)$) positive function that increases on both sides. For intuition, one can think $\psi(x)$ behaves roughly like $|x|$.

### B.3.1 PROOF OF CORRECTNESS

We prove the correctness of Theorem 3.3, we will defer some simple calculations into later sections.

**Definition of $x$ and $y$.** Consider the $k$-th round of the algorithm. For all $i \in [n]$, we define $x_i^{(k)} \in \mathbb{R}^d$, $x_i^{(k+1)} \in \mathbb{R}^d$ and $y_i^{(k)} \in \mathbb{R}^d$ as follows:

$$x_i^{(k)} = w_i^{(k)} - v_i^{(k)}, y_i^{(k)} = w_i^{(k+1)} - v_i^{(k)}, x_i^{(k+1)} = w_i^{(k+1)} - v_i^{(k+1)}.$$

Note that the difference between $x_i^{(k)}$ and $y_i^{(k)}$ is that $w$ is changing. The difference between $y_i^{(k)}$ and $x_i^{(k+1)}$ is that $v$ is changing.

**Assume sorting.** Assume the coordinates of vector $x^{(k)} \in \mathbb{R}^{n \times d}$ are sorted such that $\|x_i^{(k)}\|_2 \geq \|x_{i+1}^{(k)}\|_2$. Let $\tau$ and $\pi$ are permutations such that $\|x_{\tau(i)}^{(k+1)}\|_2 \geq \|x_{\tau(i+1)}^{(k+1)}\|_2$ and $\|y_{\pi(i)}^{(k)}\|_2 \geq \|y_{\pi(i+1)}^{(k)}\|_2$.

**Definition of Potential function.** Let $\psi : \mathbb{R} \to \mathbb{R}$ be defined by

$$\psi(x) = \begin{cases} \frac{\|x\|_2^2}{2\epsilon_{\mathrm{mds}}}, & \|x\|_2 \in [0, \epsilon_{\mathrm{mds}}] \\ \epsilon_{\mathrm{mds}} - \frac{(4\epsilon_{\mathrm{mds}}^2 - \|x\|_2^2)^2}{18\epsilon_{\mathrm{mds}}^3}, & \|x\|_2 \in (\epsilon_{\mathrm{mds}}, 2\epsilon_{\mathrm{mds}}] \\ \epsilon_{\mathrm{mds}}, & \|x\|_2 \in (2\epsilon_{\mathrm{mds}}, +\infty) \end{cases} \tag{5}$$

We define the potential at the $k$-th round by

$$\Psi_k = \sum_{i=1}^n g_i \cdot \psi(x_{\tau_k(i)}^{(k)}),$$

where $\tau_k(i)$ is the permutation such that $\|x_{\tau_k(i)}^{(k)}\|_2 \geq \|x_{\tau_k(i+1)}^{(k)}\|_2$.

**Bounding the potential.**

We can express $\Psi_{k+1} - \Psi_k$ as follows:

$$\Psi_{k+1} - \Psi_k = \sum_{i=1}^n g_i \cdot \left( \psi(x_{\tau(i)}^{(k+1)}) - \psi(x_i^{(k)}) \right)$$

$$= \sum_{i=1}^n g_i \cdot \underbrace{\left( \psi(y_{\pi(i)}^{(k)}) - \psi(x_i^{(k)}) \right)}_{w \text{ move}} - \sum_{i=1}^n g_i \cdot \underbrace{\left( \psi(y_{\pi(i)}^{(k)}) - \psi(x_{\tau(i)}^{(k+1)}) \right)}_{v \text{ move}}. \tag{6}$$

Now, using Lemma B.5 and B.7, and the fact that $\Psi_0 = 0$ and $\Psi_T \geq 0$, with Eq. 6, we get

$$0 \leq \Psi_T - \Psi_0 = \sum_{k=0}^{T-1} (\Psi_{k+1} - \Psi_k)$$

$$\leq \sum_{k=0}^{T-1} \left( O(C_1 + C_2/\epsilon_{\mathrm{mds}}) \cdot \|g\|_2 - \Omega(\epsilon_{\mathrm{mds}} r_k g_{r_k} / \log n) \right)$$

$$= T \cdot O(C_1 + C_2/\epsilon_{\mathrm{mds}}) \cdot \|g\|_2 - \sum_{k=1}^{T} \Omega(\epsilon_{\mathrm{mds}} r_k g_{r_k} / \log n) \,,$$

where the third step follows by Lemma B.5 and Lemma B.7 and $r_k$ is the number of coordinates we update during that iteration.

Therefore, we get,

$$\sum_{k=1}^{T} r_k g_{r_k} = O \left( T \cdot (C_1/\epsilon_{\mathrm{mds}} + C_2/\epsilon_{\mathrm{mds}}^2) \cdot \log n \cdot \|g\|_2 \right).$$

### B.3.2 Initialization time, update time, query time

To formalize the amortized runtime proof, we first analyze the initialization time (Lemma B.2), update time (Lemma B.3), and query time (Lemma B.4) of our maintenance data-structure.

**Lemma B.2** (Initialization time). *The initialization time of data-structure* MAINTAIN *(Algorithm 1) is* $O(dn^{1+\rho})$.

**Lemma B.3** (Update time). *The update time of data-structure* MAINTAIN *(Algorithm 1) is* $O(r g_r) + O(Sd + S \log n)$ *where* $r$ *is the number of indices we updated in* $v$, $S$ *is the number of weights changed. The later two terms are dominated by the updating time of neural network.*

*Proof.* We change $r$ ($r \geq n^a$) terms each time, and the running time for updating the LSH is $r g_r$. We need to update $v$ and calculate its $\ell_2$ norm every time, and the computational cost is $Sd$. We also need to maintain an order on the error $y_i^{(k)}$, using some standard data structure (like Fibonacci heap), this can be done in $S \log n$. Thus the total running time per update is $O(r g_r) + O(Sd + S \log n)$, the second term is bounded by the updating cost of neural network training, since calculate the gradient takes at least $O(Sd)$ and we usually have $d \gg \log n$. $\qquad \square$

**Lemma B.4** (Query time). *The query time of data-structure* MAINTAIN *(Algorithm 1) is* $O(n^\rho d + n^a d)$.

### B.4 Bounding $w$ move

**Lemma B.5** ($w$ move). *We have*

$$\sum_{i=1}^{n} g_i \cdot \mathbb{E} \left[ \psi(y_{\pi(i)}^{(k)}) - \psi(x_i^{(k)}) \right] \leq O(C_1 + C_2/\epsilon_{\mathrm{mds}}) \cdot \|g\|_2.$$

*Proof.* Observe that since the errors $\|x_i^{(k)}\|_2$ are sorted in descending order, and $\psi(x)$ is symmetric and non-decreasing function, thus $\psi(x_i^{(k)})$ is also in decreasing order. In addition, note that $g$ is decreasing, we have

$$\sum_{i=1}^{n} g_i \psi(x_{\pi(i)}^{(k)}) \leq \sum_{i=1}^{n} g_i \psi(x_i^{(k)}). \qquad (7)$$

Hence we have

$$\mathbb{E}\left[\sum_{i=1}^{n} g_i \cdot \left(\psi(y_{\pi(i)}^{(k)}) - \psi(x_i^{(k)})\right)\right] \leq \mathbb{E}\left[\sum_{i=1}^{n} g_i \cdot \left(\psi(y_{\pi(i)}^{(k)}) - \psi(x_{\pi(i)}^{(k)})\right)\right] \qquad \text{by Eq. 7}$$

$$= \sum_{i=1}^{n} g_i \cdot \mathbb{E}[\psi(y_{\pi(i)}^{(k)}) - \psi(x_{\pi(i)}^{(k)})]$$

$$= O(C_1 + C_2/\epsilon_{\mathrm{mds}}) \cdot \|g\|_2. \qquad \text{by Lemma B.6}$$

Thus, we complete the proof of $w$ move Lemma. $\qquad\square$

It remains to prove the following Lemma,

**Lemma B.6.**

$$\sum_{i=1}^{n} g_i \cdot \mathbb{E}[\psi(y_{\pi(i)}^{(k)}) - \psi(x_{\pi(i)}^{(k)})] = O(C_1 + C_2/\epsilon_{\mathrm{mds}}) \cdot \|g\|_2.$$

*Proof.* We separate the term into two:

$$\sum_{i=1}^{n} g_i \cdot \mathbb{E}[\psi(y_{\pi(i)}^{(k)}) - \psi(x_{\pi(i)}^{(k)})] = \sum_{i=1}^{n} g_{\pi^{-1}(i)} \cdot \mathbb{E}[\psi(y_i^{(k)}) - \psi(\mathbb{E}[y_i^{(k)}])]$$

$$+ \sum_{i=1}^{n} g_{\pi^{-1}(i)} \cdot (\psi(\mathbb{E}[y_i^{(k)}]) - \psi(x_i^{(k)})). \qquad (8)$$

For the first term, we have

$$\psi(y_i^{(k)}) - \mathbb{E}[\psi(y_i^{(k)})]$$

$$= \langle \psi'(\mathbb{E}[y_i^{(k)}]), y_i^{(k)} - \mathbb{E}[y_i^{(k)}]\rangle + \frac{1}{2}(y_i^{(k)} - \mathbb{E}[y_i^{(k)}])^\top \psi''(\zeta)(y_i^{(k)} - \mathbb{E}[y_i^{(k)}])$$

$$\leq \langle \psi'(\mathbb{E}[y_i^{(k)}]), y_i^{(k)} - \mathbb{E}[y_i^{(k)}]\rangle + \frac{1}{2}L_2\|y_i^{(k)} - \mathbb{E}[y_i^{(k)}]\|_2^2$$

$$= \langle \psi'(\mathbb{E}[y_i^{(k)}]), w_i^{(k+1)} - \mathbb{E}[w_i^{(k+1)}]\rangle + \frac{1}{2}L_2\|w_i^{(k+1)} - \mathbb{E}[w_i^{(k+1)}]\|_2^2 \qquad (9)$$

where the first step follows from the Mean value theorem, the second step follows from the definition of $L_2$ (see Part 4 of potential lemma B.8), the last step follows from the definition of $y_i^{(k)}$.

Next, we denote $\gamma_i = \mathrm{Var}[w_i^{(k+1)}]$. Summing over $i$ and taking conditional expectation given $w^{(k)}$ on both sides, we get

$$\sum_{i=1}^{n} g_{\pi^{-1}(i)} \cdot \mathbb{E}[\psi(y_i^{(k)}) - \psi(\mathbb{E}[y_i^{(k)}])] \leq \sum_{i=1}^{n} g_{\pi^{-1}(i)} \cdot \mathbb{E}[\langle \psi'(\mathbb{E}[y_i^{(k)}]), w_i^{(k+1)} - \mathbb{E}[w_i^{(k+1)}]\rangle]$$

$$+ \sum_{i=1}^{n} g_{\pi^{-1}(i)} \cdot \frac{1}{2}L_2\mathbb{E}[\|w_i^{(k+1)} - \mathbb{E}[w_i^{(k+1)}]\|_2^2]$$

$$= 0 + \frac{1}{2}L_2\sum_{i=1}^{n} g_{\pi^{-1}(i)} \cdot \gamma_i$$

$$\leq \frac{1}{2}L_2\|g\|_2(\sum_{i=1}^{n} \gamma_i^2)^{\frac{1}{2}}$$

$$\leq \frac{1}{2} \cdot O(1/\epsilon_{\mathrm{mds}}) \cdot \|g\|_2 \cdot C_2$$

$$= O(C_2/\epsilon_{\mathrm{mds}})\|g\|_2. \qquad (10)$$

The first step follows from Eq. 9, the second step follows from the linearity of expectation and the definition of $\gamma_i$, the third step follows from Cauchy-Shwarz inequality, the fourth step follows from $L_2 = O(1/\epsilon_{\mathrm{mds}})$ (see Lemma B.8) and Eq. 1.

For the second term, conditioning on $w_i^{(k)}$, we have

$$\psi(\mathbb{E}[y_i^{(k)}]) - \psi(x_i^{(k)}) \leq L_1 \cdot \|\mathbb{E}[y_i^{(k)}] - x_i^k\|_2 = L_1 \cdot \|\mathbb{E}[w_i^{(k+1)}] - w_i^k\|_2 \stackrel{\text{def}}{=} L_1 \cdot \beta_i. \qquad (11)$$

The first inequality follows from the $L_1$-Lipschitz continuity of $\Psi$ (see part 4 of Lemma B.8). The second step follows from the definition of $y_i^{(k)}$ and $x_i^{(k)}$.

Summing over $i$, we get

$$\sum_{i=1}^n g_{\pi^{-1}(i)} \cdot (\psi(\mathbb{E}[y_i^{(k)}]) - \psi(x_i^{(k)})) \leq \sum_{i=1}^n g_{\pi^{-1}(i)} \cdot L_1 \beta_i \leq L_1 \cdot \|g\|_2 \cdot (\sum_{i=1}^n \beta_i^2)^{\frac{1}{2}} \leq O(C_1) \cdot \|g\|_2. \tag{12}$$

The first step follows from Eq. 11, the second step follows from Cauchy Shwarz inequality, the last step follows from $L_1 \leq 2$ (see part 4 of Lemma B.8) and Eq. 1.

Combining Eq. 81012, we have

$$\sum_{i=1}^n g_i \cdot \mathbb{E}[\psi(y_{\pi(i)}^{(k)}) - \psi(x_{\pi(i)}^{(k)})] \leq O(C_1 + C_2/\epsilon_{\text{mds}})\|g\|_2.$$

Thus completing the proof. $\qquad\qquad\square$

## B.5 Bounding $v$ move

The goal of this section is to prove Lemma B.7.

**Lemma B.7** ($v$ move). *We have,*

$$\sum_{i=1}^n g_i \cdot \left( \psi(y_{\pi(i)}^{(k)}) - \psi(x_{\tau(i)}^{(k+1)}) \right) \geq \Omega(\epsilon_{\text{mds}} r_k g_{r_k}/ \log n).$$

*Proof.* We split the proof into two cases.

We first understand some simple facts which are useful in the later proof. Note that from definition of $x_i^{(k+1)}$, we know that $x^{(k+1)}$ has $r_k$ coordinates are $\vec{0}$. Basically, $\|y^{(k)} - x^{(k+1)}\|_0 = r_k$. The difference between those vectors is, for the largest $r_k$ coordinates in $y^{(k)}$, we erase them in $x^{(k+1)}$. Then for each $i \in [n - r_k]$, $x_{\tau(i)}^{(k+1)} = y_{\pi(i+r_k)}^{(k)}$. For convenience, we define $y_{\pi(n+i)}^{(k)} = \vec{0}, \forall i \in [r_k]$.

**Case 1.** We exit the while loop when $1.5 r_k \geq n$.

Let $u^*$ denote the largest $u$ satisfying $\|y_{\pi(u)}^{(k)}\|_2 \geq \epsilon_{\text{mds}}/2$. If $u^* \geq r_k$, then we have that $\|y_{\pi(r_k)}^{(k)}\|_2 \geq \epsilon_{\text{mds}}/2 \geq \epsilon_{\text{mds}}/100$. Otherwise, the condition of the loop shows that

$$\begin{aligned}
\|y_{\pi(r_k)}^{(k)}\|_2 &\geq (1 - 1/\log n)^{\log_{1.5} r_k - \log_{1.5} u^*} \|y_{\pi(u^*)}^{(k)}\|_2 \\
&\geq (1 - 1/\log n)^{\log_{1.5} n} \epsilon_{\text{mds}}/2 \\
&\geq \epsilon_{\text{mds}}/100.
\end{aligned}$$

where we used that $n \geq 4$.

According to the definition of $x_{\tau(i)}^{(k+1)}$, we have

$$\sum_{i=1}^{n} g_i(\psi(y_{\pi(i)}^{(k)}) - \psi(x_{\tau(i)}^{(k+1)})) = \sum_{i=1}^{n} g_i(\psi(y_{\pi(i)}^{(k)}) - \psi(y_{\pi(i+r_k)}^{(k)}))$$

$$\geq \sum_{i=n/3+1}^{n} g_i(\psi(y_{\pi(i)}^{(k)}) - \psi(y_{\pi(i+r_k)}^{(k)}))$$

$$\geq \sum_{i=n/3+1}^{n} g_i \psi(y_{\pi(i)}^{(k)})$$

$$\geq \sum_{i=n/3+1}^{2n/3} g_i f((\epsilon_{\text{mds}}/100)^2) \geq \Omega(r_k g_{r_k} \epsilon_{\text{mds}}),$$

where the first step follows from $x_{\tau(i)}^{(k+1)} = y_{\pi(i+r_k)}^{(k)}$, the second step follows from $\psi(x)$ is non-decreasing (see part 2 of Lemma B.8) and $\|y_{\pi(i)}^{(k)}\|_2$ is non-increasing, the third step follows from $1.5r_k > n$ and hence $\psi(y_{\pi(i+r_k)}^{(k)}) = 0$ for $i \geq n/3+1$, the fourth step follows from $\psi(x) = f(\|x\|_2^2)$ is non-decreasing and $\|y_{\pi(i)}^{(k)}\|_2 \geq \|y_{\pi(r_k)}^{(k)}\|_2 \geq \epsilon_{\text{mds}}/100$ for all $i < 2n/3$, and the last step follows by $g$ is non-increasing and the part 3 of Lemma B.8.

**Case 2.** We exit the while loop when $1.5r_k < n$ and $\|y_{\pi(1.5r_k)}^{(k)}\|_2 < (1 - 1/\log n)\|y_{\pi(r_k)}^{(k)}\|_2$.

By the same argument as Case 1, we have that $\|y_{\pi(r_k)}^{(k)}\|_2 \geq \epsilon_{\text{mds}}/100$. Part 3 of Lemma B.8 together with the fact

$$\|y_{\pi(1.5r_k)}^{(k)}\|_2 < \min(\epsilon_{\text{mds}}/2, \|y_{\pi(r_k)}^{(k)}\|_2 \cdot (1 - 1/\log n)),$$

indicates that

$$\psi(y_{\pi(1.5r_k)}^{(k)}) - \psi(y_{\pi(r_k)}^{(k)}) = \Omega(\epsilon_{\text{mds}}/\log n). \tag{13}$$

Putting things together, we have

$$\sum_{i=1}^{n} g_i \cdot (\psi(y_{\pi(i)}^{(k)}) - \psi(x_{\tau(i)}^{(k+1)}))$$

$$= \sum_{i=1}^{n} g_i \cdot (\psi(y_{\pi(i)}^{(k)}) - \psi(y_{\pi(i+r_k)}^{(k)})) \qquad \text{by } x_{\tau(i)}^{(k+1)} = y_{\pi(i+r_k)}^{(k)}$$

$$\geq \sum_{i=r_k/2}^{r_k} g_i \cdot (\psi(y_{\pi(i)}^{(k)}) - \psi(y_{\pi(i+r_k)}^{(k)})) \qquad \text{by } \psi(y_{\pi(i)}^{(k)}) - \psi(y_{\pi(i+r_k)}^{(k)}) \geq 0$$

$$\geq \sum_{i=r_k/2}^{r_k} g_i \cdot (\psi(y_{\pi(r_k)}^{(k)}) - \psi(y_{\pi(1.5r_k)}^{(k)}))$$

$$\geq \sum_{i=r_k/2}^{r_k} g_i \cdot \Omega(\frac{\epsilon_{\text{mds}}}{\log n}) \qquad \text{by } 13$$

$$\geq \sum_{i=r_k/2}^{r_k} g_{r_k} \cdot \Omega(\frac{\epsilon_{\text{mds}}}{\log n}) \qquad \text{by } g_i \text{ is decreasing}$$

$$= \Omega\left(\epsilon_{\text{mds}} r_k g_{r_k}/\log n\right),$$

where the third step follows by $\|y_{\pi(i)}^{(k)}\|_2$ is non-increasing and $\psi$ is non-decreasing (see part 2 of Lemma B.8). $\qquad \square$

### B.6 POTENTIAL FUNCTION $\psi$

**Lemma B.8** (Properties of potential function $\psi$). *Let function $\psi : \mathbb{R}^d \to \mathbb{R}$ (defined in Eq. 5). Then function $\psi$ satisfies the following properties:*
*1. Symmetric ($\psi(-x) = \psi(x)$) and $\psi(0) = 0$;*
*2. If $\|x\|_2 \geq \|y\|_2$, then $\psi(x) \geq \psi(y)$;*
*3. $|f'(x)| = \Omega(1/\epsilon_{\mathrm{mds}}), \forall |x| \in [(0.01\epsilon_{\mathrm{mds}})^2, \epsilon_{\mathrm{mds}}^2]$;*
*4. $L_1 \stackrel{\mathrm{def}}{=} \max_x \frac{D_x \psi[h]}{\|h\|_2} = 2$ and $L_2 \stackrel{\mathrm{def}}{=} \max_x \frac{D_x^2 \psi[h,h]}{\|h\|_2^2} = 10/\epsilon_{\mathrm{mds}}$;*
*5. $\psi(x)$ is a constant for $\|x\|_2 \geq 2\epsilon_{\mathrm{mds}}$*

*Proof.* Recall $f : \mathbb{R}_+ \to \mathbb{R}$ is defined as

$$f(x) := \begin{cases} \frac{x^2}{2\epsilon_{\mathrm{mds}}^3}, & x \in [0, \epsilon_{\mathrm{mds}}^2]; \\ \epsilon_{\mathrm{mds}} - \frac{(4\epsilon_{\mathrm{mds}}^2 - x)^2}{18\epsilon_{\mathrm{mds}}^3}, & x \in (\epsilon_{\mathrm{mds}}^2, 4\epsilon_{\mathrm{mds}}^2]; \\ \epsilon_{\mathrm{mds}}, & x \in (4\epsilon_{\mathrm{mds}}^2, +\infty). \end{cases}$$

We can see that

$$f(x)' = \begin{cases} \frac{x}{\epsilon_{\mathrm{mds}}^3}, & x \in [0, \epsilon_{\mathrm{mds}}^2]; \\ \frac{4\epsilon_{\mathrm{mds}}^2 - x}{9\epsilon_{\mathrm{mds}}^3}, & x \in (\epsilon_{\mathrm{mds}}^2, 4\epsilon_{\mathrm{mds}}^2]; \\ 0, & x \in (4\epsilon_{\mathrm{mds}}^2, +\infty). \end{cases}$$

and

$$f(x)'' = \begin{cases} \frac{1}{\epsilon_{\mathrm{mds}}^3}, & x \in [0, \epsilon_{\mathrm{mds}}^2]; \\ -\frac{1}{9\epsilon_{\mathrm{mds}}^3}, & x \in (\epsilon_{\mathrm{mds}}^2, 4\epsilon_{\mathrm{mds}}^2]; \\ 0, & |x| \in (4\epsilon_{\mathrm{mds}}^2, +\infty) \end{cases}$$

It implies that $\max_x |f(x)'| \leq \frac{1}{\epsilon_{\mathrm{mds}}}$ and $\max_x |f(x)''| \leq \frac{1}{\epsilon_{\mathrm{mds}}^3}$. Let $\psi(x) = f(\|x\|_2^2)$.

**Proof of Part 1, 2 and 5.** These proofs are pretty standard from definition of $\psi$.

**Proof of Part 3.** This is trivially following from definition of scalar function $f$.

**Proof of Part 4.** By chain rule, we have
$$D_x \psi[h] = 2f'(\|x\|_2^2) \cdot \langle x, h \rangle$$
$$D_x^2 \psi[h, h] = 2f''(\|x\|_2^2) \cdot (\langle x, h \rangle)^2 + 2f'(\|x\|_2^2) \cdot \langle h, h \rangle$$
We can upper bound
$$|D_x \psi[h]| \leq 2|f'(\|x\|_2^2)| \cdot |\langle x, h \rangle| \leq 2|f'(\|x\|_2^2)| \cdot \|x\|_2 \cdot \|h\|_2.$$
Thus, we have
$$|f'(\|x\|_2^2)| \cdot \|x\|_2 = \begin{cases} \|x\|_2^3/\epsilon_{\mathrm{mds}}^3 \leq 1, & \|x\|_2 \in [0, \epsilon_{\mathrm{mds}}]; \\ (4\epsilon_{\mathrm{mds}}^2 - \|x\|_2^2)\|x\|_2/(9\epsilon_{\mathrm{mds}}^3) \leq 2/3, & \|x\|_2 \in (\epsilon_{\mathrm{mds}}, 2\epsilon_{\mathrm{mds}}); \\ 0, & \|x\|_2 \in (2\epsilon_{\mathrm{mds}}, +\infty). \end{cases}$$
It implies that $|D_x \psi[h]| \leq 2\|h\|_2, \forall x$.

By case analysis, we have
$$|f''(\|x\|_2^2)| \cdot \|x\|_2^2 \leq \begin{cases} \frac{1}{\epsilon_{\mathrm{mds}}^3}\|x\|_2^2 \leq 4/\epsilon_{\mathrm{mds}}, & \|x\|_2^2 \in [0, 4\epsilon_{\mathrm{mds}}^2]; \\ 0, & \|x\|_2^2 \in (4\epsilon_{\mathrm{mds}}, +\infty). \end{cases}$$
We can also upper bound
$$\begin{aligned} |D_x^2 \psi[h, h]| &\leq 2|f''(\|x\|_2^2)| \cdot \langle x, h \rangle^2 + 2|f'(\|x\|_2^2)| \cdot \|h\|_2^2 \\ &\leq 2|f''(\|x\|_2^2)| \cdot (\|x\|_2 \cdot \|h\|_2)^2 + 2|f'(\|x\|_2^2)| \cdot \|h\|_2^2 \\ &\leq 2 \cdot \frac{4}{\epsilon_{\mathrm{mds}}} \|h\|_2^2 + 2 \cdot \frac{1}{\epsilon_{\mathrm{mds}}} \|h\|_2^2 \\ &= \frac{10}{\epsilon_{\mathrm{mds}}} \|h\|_2^2. \end{aligned}$$

$\square$

### B.7 EXAMPLES

**Comparing with sequential updating algorithm**   We present another version of our main result, which is particularly useful when we compare with the sequential updating algorithm.

**Theorem B.9** (Locality sensitive hashing maintenance, worst case bound). *For any constant $c_1, c_2$ ($c_1 > c_2$) there is a dynamic data structure (Algorithm 1) that achieves $(c_1, c_2)$-accuracy. The data structure takes $\widetilde{O}(dn^{1+\rho})$ time to initialize and each call of $\mathrm{QUERY}(h)$ takes time $\widetilde{O}(n^\rho d)$. By taking $a = \min\{\rho, \alpha\}$ and $g_r = t_r$, after $T$ iterations, the amortized expected time per call of $\mathrm{UPDATE}(w)$ is at most*

$$\widetilde{O}\left(\frac{n^\rho}{T}\sum_{k=0}^{T-1}\sum_{i=1}^{n}\left(\left|\mathbb{E}[w_i^{(k+1)}] - w_i^{(k)}\right| + \mathrm{Var}[w_i^{k+1}]\right)\right) \tag{14}$$

The proof is almost indentical to Theorem B.1, we only need to replace Lemma B.5 by the following one.

**Lemma B.10** ($w$ move). *We have*

$$\sum_{i=1}^{n} g_i \cdot \mathbb{E}\left[\psi(y_{\pi(i)}^{(k)}) - \psi(x_i^{(k)})\right] \leq O\left(g_1 \sum_{i=1}^{n}\left|\mathbb{E}[w_i^{(k+1)}] - w_i^{(k)}\right| + \mathrm{Var}[w_i^{k+1}]\right)$$

$$= O\left(n^\rho \sum_{i=1}^{n}\left|\mathbb{E}[w_i^{(k+1)}] - w_i^{(k)}\right| + \mathrm{Var}[w_i^{k+1}]\right)$$

*We omit the dependence on $\epsilon_{\mathrm{mds}}$ since we assume it to be constant in the scheduler.*

*Proof.* Again, the proof is almost identical to Lemma B.5, the only difference is that we use $\ell_\infty/\ell_1$ form of Cauchy-Shwarz instead of the $\ell_2/\ell_2$ form in the previous proof. We only point out the difference here.

We replace Eq. 10 with

$$\sum_{i=1}^{n} g_{\pi^{-1}(i)} \cdot \mathbb{E}[\psi(y_i^{(k)}) - \psi(\mathbb{E}[y_i^{(k)}])] = 0 + \frac{1}{2}L_2 \sum_{i=1}^{n} g_{\pi^{-1}(i)} \cdot \gamma_i$$

$$\leq \frac{1}{2}L_2 g_1 \cdot \sum_{i=1}^{n} \gamma_i$$

$$\approx O\left(g_1 \sum_{i=1}^{n} \mathrm{Var}[w_i^{k+1}]\right).$$

The first step follows from Eq. 10, the second step follows from $\{g_i\}_{i=1,\cdots,n}$ are decreasing and the last step follows from the definition of $\gamma_i$.

Similarly, we replace Eq. 12 with

$$\sum_{i=1}^{n} g_{\pi^{-1}(i)} \cdot (\psi(\mathbb{E}[y_i^{(k)}]) - \psi(x_i^{(k)})) \leq \sum_{i=1}^{n} g_{\pi^{-1}(i)} \cdot L_1 \beta_i \lesssim O\left(g_1 \sum_{i=1}^{n}\left|\mathbb{E}[w_i^{(k+1)}] - w_i^{(k)}\right|\right)$$

The first step follows from Eq. 12, the second step follows from the definition of $\beta_i$, $L_1 \leq 2$ and $\{g_i\}_{i=1,\cdots,n}$ are decreasing.

The rest of the proof are the same and we omit it here. $\square$

The sequential updating time for LSH is $n^\rho$. The total number of updating is (roughly) at least

$$O\left(\sum_{k=1}^{T}\sum_{i=1}^{n}\left(\left|\mathbb{E}[w_i^{(k+1)}] - w_i^{(k)}\right| + \mathrm{Var}[w_i^{k+1}]\right)\right)$$

It then easy to see that our scheduler always perform better than naive sequential update.

**Comparing with naive batch updating algorithm**    Below we give a concrete example to show the effectiveness of our algorithm. We take $\alpha = \rho$ for simplicity. Remember in Assumption 3.2, we already assume for any $r \leq n^a = n^\rho$, $T_r = n^\rho$, i.e., the average LSH updating time $t_r = n^\rho / r$ decays linearly in $r$ when $r \leq n^\rho$. It remains to specify the rest $t_r$ ($r \geq n^\rho$).

**Example B.11.** *Assuming the average running time decays faster than $1/\sqrt{r}$, i.e. $t_r = n^{-(1-\beta)\rho} r^{-\beta}$ for some $\beta \in [\frac{1}{2}, 1)$ when $r \geq n^\rho$. We then have*

$$\|g\|_2^2 = \sum_{r=1}^{n} g_r^2 \approx n^{-\rho/2}.$$

*The running time of our scheduler is then at most*

$$(C_1 + C_2)\|g\|_2 \approx (C_1 + C_2)n^{\rho/2}.$$

*As long as the weight changes slowly, say $C_1 \approx C_2 \leq n^{\rho/2}$, we know the amortized updating time of our scheduler is strictly better than the naive approach that updates LSH every time, which has running time $n^\rho$ each iteration.*

## C LEARNABLE LSH

### C.1 LEARNING HASH FUNCTIONS

Denote a set of neurons in a particular layer as $\mathcal{C} = \{v_r \mid 0 \leq r < m\}$, where each neuron $v_r$ has weight and bias. Given an input embedding $q_i$ from the previous hidden layer, the output of a forward pass through neuron $r$ is $\sigma(\langle q_i, v_r \rangle)$, where $\sigma$ is some activation functions. For each input embedding $q_i$, both SLIDE and REFORMER select a set of neurons, denoted as $\mathcal{S}_i$, from the LSH hash tables for the forward pass. We collect the training samples from $q_i$ and its $\mathcal{S}_i$ to improve the performance of LSH. Formally, the pairwise training samples $(q, v)$ are collected according to the following criterion:

- positive pairs $\mathcal{P}_+ = (q, v)$ if $v \in S$ and $\langle q, v \rangle > t_+$
- negative pairs $\mathcal{P}_- = (q, v)$ if $v \in S$ and $\langle q, v \rangle < t_-$

And the loss is defined as:

$$\mathcal{L}(\mathcal{H}, \mathcal{P}_+, \mathcal{P}_-) = \max \left\{ 0, \sum_{(q,v) \in \mathcal{P}_+} - \cos(\mathcal{H}(q), \mathcal{H}(v)) + \sum_{(q,v) \in \mathcal{P}_-} \cos(\mathcal{H}(q), \mathcal{H}(v)) + \alpha \right\}. \quad (15)$$

Here $\mathcal{H}$ is the hash functions of all $L$ hash tables. $\mathcal{H}(x)$ generates a $K \cdot L$ vector containing the projection of $x$ in each hash table. $\cos(\mathcal{H}(x), \mathcal{H}(y))$ represents the cosine similarity of two projected vectors. The major contribution of this learning approach is that it optimizes the hash functions and the hash table indices together. Our method focuses on learning a useful indexing scheme for retrieval efficiency while other learnable hash methods (Wang et al., 2017) focus on binary sketches that have higher precision.

We present a demonstration of our loss in Figure 9. Two random hash functions separate 6 data points into four parts. It is obvious that point 1 (pink) should be partitioned together with point 2 instead of point 5 and 6. On the other hand, point 3 and 4 should not be separated. Therefore, during the hash function training, we choose point (5,6) as a positive pair. Meanwhile, we chose point (5,1) and (5,4) as negative pairs.

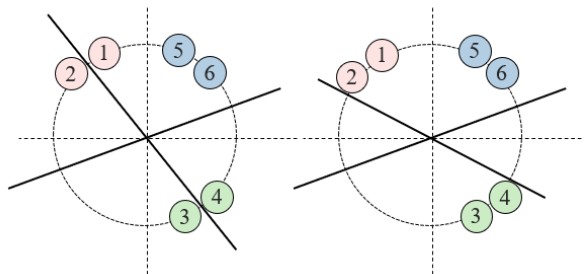

Figure 9: Illustration of learning hash functions

Then, the loss would update the hash functions to accommodate these new partitions.

## C.2 OBSERVATIONS ON ATTENTION DISTRIBUTION

In this section, we present the visualization with analysis on the distribution of the minimal quantifies of neurons that sum up to have 0.9 softmax values in attention. On Figure 10, we present the distribution of each head of attention in each layer from a transformer model trained on Enwiki8 dataset. We classify the patterns(Ramsauer et al., 2020) of the distribution into 3 categories by their median values. With this observation, we are able to determine the layers that LSH or learnable LSH can apply in MONGOOSE framework.

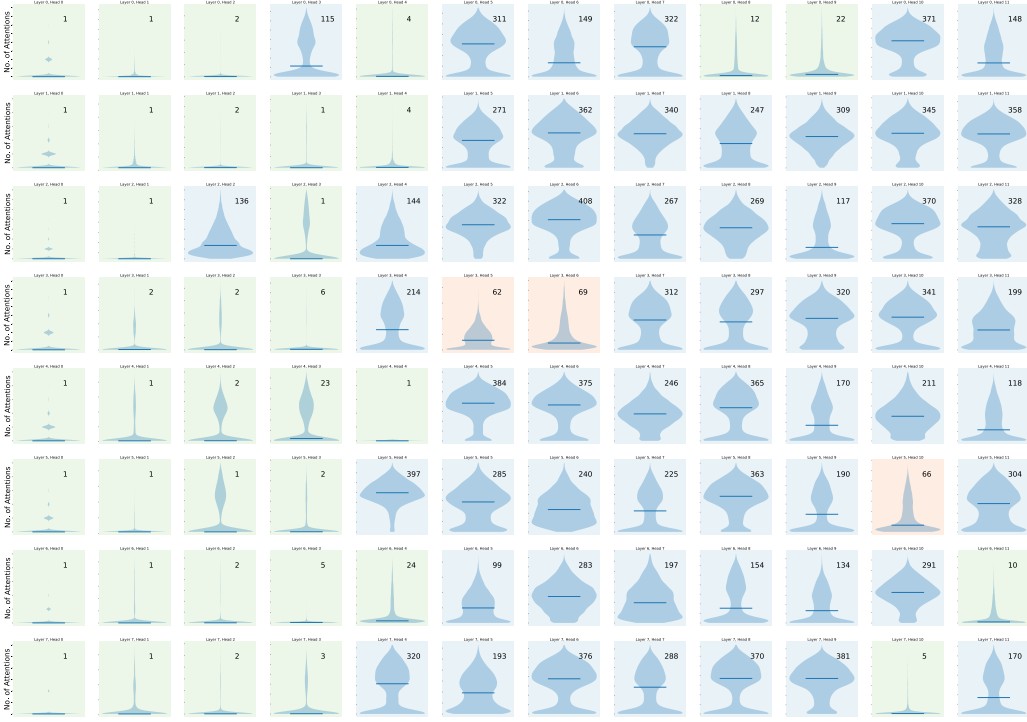

Figure 10: The distribution of the minimal quantifies of neurons that sum up to have 0.9 softmax values in attention in each head of attention in each layer of transformer for Enwiki8

# D EXPERIMENTS DETAILS

## D.1 DATA STATISTICS

Table 5: Statistics for our benchmark dataset

| Dataset | Wiki10-31k | Delicious-200K | Wiki-325K | Amz-670K |
|---|---|---|---|---|
| Output Dimension | 30938 | 205443 | 325056 | 670091 |
| Input Dimension | 101938 | 782585 | 1617899 | 135909 |
| Training Samples | 14146 | 6616 | 1778351 | 490449 |
| Testing Samples | 196606 | 100095 | 587084 | 153025 |

We present statistics on the 3 datasets we test on from the Extreme Classification Repository (Bhatia et al., 2016). While the number of datapoints in each dataset is not large (on the order of 200K at most), the key feature is the sheer size of the input and output dimensions. In particular, each dataset has over 10,000 output classes, which, using a conventional nueral network, requires a matrix multiplication involving over 10,000 neurons at the final layer.

## D.2 INFLUENCE OF THE PARAMETERS

**Smart Scheduler Parameters:** Recall that in Algorithm 1, $\epsilon_{\mathrm{mds}}$ is the key parameter to our smart scheduler because the number of neurons exceeding this threshold determines if LSH updates are needed. In practice, instead of choosing a single $\epsilon_{\mathrm{mds}}$, we use some fraction of the $\ell_2$ norm of the weights as the update threshold. We empirically observe from the previous experiments that MONGOOSE is robust to any choice of the fraction in the range 0.1-0.3. Thus, it can consistently outperform other baselines without heavily tuning this parameter for different datasets.

**Learnable LSH Parameters:** For LSH-based NNS-ds, the number of hashes is one key parameter to balance NNS quality and efficiency. In Table 6 we study the effects of varying the number of hashes in MONGOOSE. We observe that compared to our best empirical choice (10 hashes), both a smaller (8) and larger number (12) of hashes increases convergence

Table 6: MONGOOSE performance as a function of number of hashes.

| Number of Hashes | $P@1$ | $P@5$ | Speed (batch/s) |
|---|---|---|---|
| 8 | 0.492 | 0.241 | 5.79 |
| 10 | 0.519 | 0.255 | 7.7 |
| 12 | 0.469 | 0.234 | 9.92 |

time. MONGOOSE with fewer hashes selects more unnecessary neurons and lowers training efficiency. A larger number of hashes might miss necessary neurons with higher probabilities and would similarly take longer to converge. Another important parameter is the threshold for positive and negative examples to train hash functions. In our experiments, top-1 and bottom-1 are the most effective choices (top-3 and top-5 do not make much difference). In general, this threshold in our design is robust and does not require heavy tuning.

## D.3 ADDITIONAL RESULTS ON THE EXTREME-CLASSIFICATION TASK

We present the results of additional experiments comparing the classification performance of MONGOOSE against SLIDE and FULL in Figure 11. The two metrics, P@1 and P@5, are presented in (Bhatia et al., 2016) as follows:

$$\mathrm{P}@k = \frac{1}{k} \sum_{l \in \mathrm{rank}_k(\widehat{\mathbf{y}})} \mathbf{y}_l$$

where $\mathrm{rank}_k(\cdot)$ extracts the top $k$ indices from a vector, $\widehat{\mathbf{y}}$ refers to the predicted labels, and $\mathbf{y}$ refers to the ground truth label.

We also present the results of time profiling to demonstrate the speedups of MONGOOSE in each layer. As shown in Table 7, $\mathrm{Time}_1$ represents the total convergence time while $\mathrm{Time}_2$ represents the execution time in the wide output layer (including the forward and backward pass). Here we observe that MONGOOSE provides up to $20\times$ speedup in the second layer. The major reason for the increase in convergence speedup ($3\times$ to $20\times$) is that the first embedding layer for high dimensional and sparse input is time and memory consuming Medini et al. (2019). Therefore, in the end-to-end result, the advantage of MONGOOSE has been diluted.

Table 7: This table summarizes the performance of MONGOOSE, SLIDE and Full-Computation implemented with PyTorch (Paszke et al., 2019). P@1 is the top-1 accuracy and P@5 is the top-5 accuracy. Time represents convergence time.

| Datasets | Full-Computation | | | | | | SLIDE | | | | | | MONGOOSE | | | | | |
|---|---|---|---|---|---|---|---|---|---|---|---|---|---|---|---|---|---|---|
| | P@1 | P@5 | Time₁ | Time₂ | Mem₁ | Mem₂ | P@1 | P@5 | Time₁ | Time₂ | Mem₁ | Mem₂ | P@1 | P@5 | Time₁ | Time₂ | Mem₁ | Mem₂ |
| Wiki10-31K | **0.824** | 0.578 | 63 | 33 | 0.3 | 0.12 | 0.824 | 0.556 | 47 (1.3 ×) | 18 (1.8×) | 0.24 (1.3×) | 0.06 (2×) | 0.824 | **0.618** | **35 (1.8×)** | **5 (6.6×)** | **0.2 (1.5×)** | **0.03 (4×)** |
| Delicious-200K | 0.446 | 0.356 | 483 | 361 | 2.2 | 0.9 | 0.465 | **0.372** | 318 (1.5×) | 197 (1.8×) | 1.7 (1.3×) | 0.4 (2.2×) | **0.469** | 0.371 | **162 (3×)** | **42 (8.6×)** | **1.5 (1.5×)** | **0.2 (4.5×)** |
| Wiki-325K | 0.501 | 0.235 | 5702 | 3990 | 3.9 | 1.5 | 0.438 | 0.205 | 4680 (1.2×) | 2970 (1.4×) | 3.3 (1.2×) | 0.9 (1.7×) | **0.519** | **0.255** | **1906 (3×)** | **200(20 ×)** | **2.7 (1.5×)** | **0.4 (4×)** |

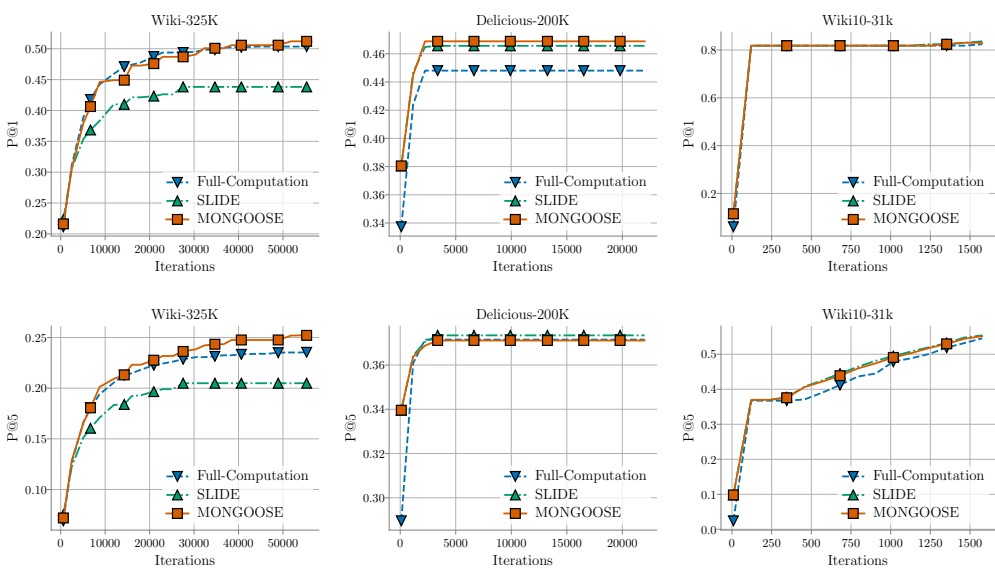

Figure 11: Comparison of MONGOOSE against SLIDE and FULL during the training. The two metrics (P@1 in the top row and P@5 in the bottom row) are the same as in (Bhatia et al., 2016).

## D.4 ADDITIONAL RESULTS ON THE LANGUAGE MODELING TASK

We present additional experiments comparing the training loss of MONGOOSE and Reformer on the synthetic copy task in Section 4. In Figure 12 we present additional experiments comparing the training loss of MONGOOSE and Reformer on the synthetic copy task in Section 4.1.2. The titles of each graph denote the hyperparameters of the Transformer model being tested: h1_s2048_t16 refers to a model with 1 round of hashing, a sequence length of 2048, and a token size of 16. Overall we can see that MONGOOSE makes a marked improvement in loss over Reformer in every case.

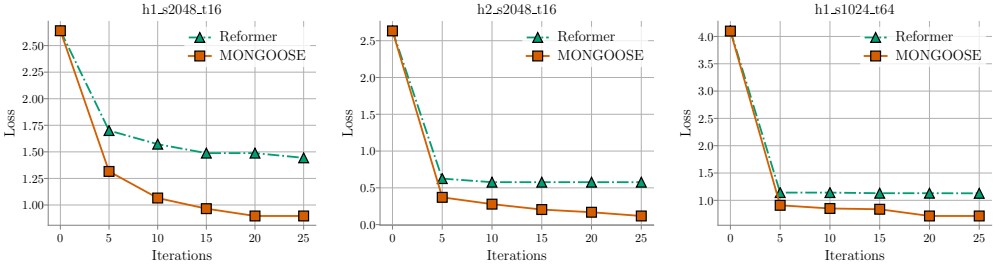

Figure 12: Comparison of MONGOOSE against Reformer during training on the synthetic copy task.

In Figure 13, we present further results comparing MONGOOSE to Reformer on the enwik8 character-level language modelling task. In particular, we train two sizes of Transformer (3 and 6 layers) with

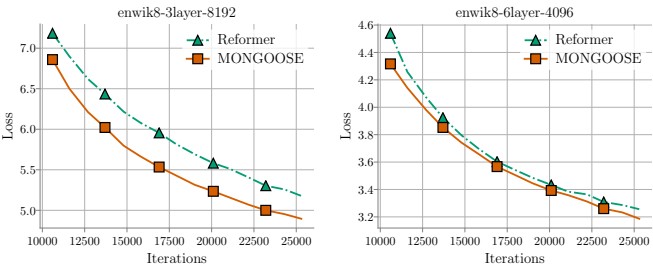

Figure 13: Comparison of MONGOOSE against Reformer during training on enwik8.

a maximum sequence length of 8192 and 4096 respectivelyh. In both settings mongoose achieves lower loss than Reformer.

Note that the goal of this experiments is to prove the superiority of learnable LSH over classical LSH in NN training (or MONGOOSE vs. naive LSH-NN framework) rather than improve the state-of-the-art perplexity. For 6 layer transformer experiment on enwik8, Reformer only reaches 2.71 in perplexity, while MONGOOSE achieves 2.65 in the same setting. Both MONGOOSE and Reformer (with LSH) save 6× memory than the original transformers in above two tasks.

# E  RELATED WORK

## E.1  DATA STRUCTURES FOR DYNAMIC SIMILARITY SEARCH

Similarity search is a well-studied problem with wide applications in recommendation system (Xue et al., 2017; Severyn & Moschitti, 2015; Hall & Attenberg, 2015), question answering (Boytsov et al., 2016; Seo et al., 2019; Ahmad et al., 2019; Chang et al., 2020), multi-label classification (Covington et al., 2016; Jain et al., 2016; Tagami, 2017) and natural language processing (Bengio et al., 2003; Gao et al., 2014; Lee et al., 2016). There are two major categories of similarity measures: (1) metric similarity (cosine similarity, euclidean distance), (2) non-metric similarity (inner product, KL divergence, neural network). The brute-force approach to solving similarity search is computationally expensive; in response, researchers have developed novel indexing structures to accelerate the search process, with trade-offs on search accuracy. Based on these indexing structures, similarity search algorithms can be broadly categorized as (1) hashing (Shrivastava & Li, 2014; 2015), (2) quantization (Guo et al., 2016; Jegou et al., 2011), (3) tree-based (Ram & Gray, 2012), or (4) graph-based methods (Malkov et al., 2012; 2014; Malkov & Yashunin, 2018).

Most similarity search scenarios studied by these papers are static (search data does not change). In the experiments section, the developed similarity search methods are compared on a fixed dataset such as Sift (Jegou et al., 2011) or Glove (Pennington et al., 2014). However, in current similarity search applications such as the work of (Fan et al., 2019), the search distribution (e.g. product vectors) changes over time due to the activation of new products and the expiration of old products. Therefore, some similarity search methods in metric space have been modified for online settings, such as hashing (Coleman et al., 2019) and quantization (Xu et al., 2018) methods; these methods sacrifice search speed for data-adaptiveness.

The training phase of Deep Learning models provides a natural setting for Dynamic LSH. During training, the weight matrices are slowly modified via gradients derived from objective functions. If we consider the weights as the search data and the training sample as queries, we can view DNN training as a Dynamic Similarity Search problem. Recent works take advantage of this view of DNN training by introducing LSH data structures to the NN training process. (Chen et al., 2020) propose an algorithm (SLIDE) that retrieves neurons with maximum inner product in each step via an LSH based data structure. In this way, the backward pass of NN training is concentrated on the neurons with estimated large gradients. Their CPU implementation is able to outperform a traditional GPU implementation. Similar hashing based algorithms have also been used in Transformer models: (Kitaev et al., 2020) (Reformer) propose an LSH structure to reduce the memory bottleneck of self-attention modules especially over long sequences in Transformer.

Since DNN weights are the search data, the distribution of the weights among different hash buckets changes throughout the training. This necessitates constantly updating the LSH data structure: failing to update the LSH data structure as the search data changes degrades its nearest-neighbor search performance, which in turn worsens the quality of the DNN approximation. In our experiments, we found that failing to update the LSH data structure in SLIDE caused a 28% decrease in top-1 accuracy.

## E.2  DATA DEPENDENT INDEXING

Data dependent hashing methods (DDH) focus on adapting hashing schemes to the data distribution. They often relate to indexing via an objective function. Most literature concentrates on hashing and quantization methods. Although previous works have achieved promising results in learning B-Trees (Kraska et al., 2018) or Lattice quantization (Sablayrolles et al., 2019), there are two major bottlenecks for DDH: (1) Theoretical insights are few applied in practice, (2) Complex index designs introduce computation overhead. In the theory of data dependent indexing, (Andoni et al., 2018) present fruitful insights on hashing-based Approximate Nearest Neighbor search on metric space. However, there is not much literature that puts these insights into practice. (Dong et al., 2019) propose a $k$-NN graph based algorithm for efficiently learning data dependent LSH in Euclidean space based on the insights given by (Andoni et al., 2018). However, their method requires precomputing a $k$-NN graph, which is computationally expensive at scale. In the experiments, (Dong et al., 2019) only compare their method with $k$-means; their performance compared to major ANN benchmarks (Erik et al., 2018) remains unknown.

Many practical applications of LSH eschew DDH methods for vanilla LSH due to the aforementioned computational bottlenecks. However, despite having sub-linear query time in theory, query complexity for even vanilla LSH is still high in practice Datar et al. (2004); Andoni et al. (2017).

When designing the dynamic data structure, we want to minimize the total running time and balance the overhead and benefit brought by the data structure. In general, the matrix multiplication and backpropagation procedures comes from neural network training, which we can not control, hence, we want to balance the accuracy-efficiency trade-offs by selecting neurons.

This is especially pronounced in the Dynamic LSH regime, when weights are evolving, besides query time, LSH updates incur a more significant overhead that harms the overall efficiency. (Chen et al., 2020) introduce a method to control update frequency, but their method requires heavy hyperparameter tuning and is not guaranteed to work for each benchmark. In contrast, our findings in the following section show that the update overhead can be significantly reduced while maintaining nearest-neighbor search quality during NN training. To our knowledge, this is the first time DDH techniques have been successfully applied to the Deep Learning setting.

### E.3 EFFICIENT NEURAL NETWORK TRAINING

The weights of a neural network dynamically change during the training process, which brings great challenges to the LSH implementation. Since the data are no longer static, we need to constantly update the hash table, which incurs extra computation cost that could harm the overall performance. We present a *dynamic data structure* to handle this issue, and our data structure achieves significant speedup over naive implementation under mild assumptions, without compromising the worst case guarantee. The dynamic data structure (shown in Algorithm 1) borrows ideas from the work of (Cohen et al., 2019; Jiang et al., 2021) (which are originally designed for linear programming), and we adapt it to the LSH setting. Our data structure generalizes several practical insights and turns practical heuristics into rigorous theory, which guarantees significant speed up under natural assumptions for training neural networks and ensure the worst guarantee at the same time. In particular, we generalize and provide the following three practical heuristics (i) only update significantly changed coordinates, (ii) batch updating LSH instead of sequential updating, (iii) using "prediction" on those "marginal" coordinates.

In this work, we primarily study two LSH based efficient NN training methods: SLIDE and RE-FORMER. SLIDE introduces LSH to select neurons in the forward pass and then only do gradient descent on the chosen neurons. The major trade-off of SLIDE is the rebuild overhead versus the accurate neuron selection. To accurately retrieve the neurons with high inner products, the hash tables are required to be updated. This rehashing and rebuilding will be the major overhead caused by SLIDE. Therefore, the SLIDE's speed over full NN training is determined by the relative magnitude of rebuild overhead compared to the saved back-propagation time. The major goal of REFORMER is to reduce the memory consumption of the transformer so that GPU based hardware can support sequence to sequence tasks with longer sequence length. REFORMER also shares the trade-off when rebuilding the hash tables for better retrieval of attention weights. Besides, learning hash functions of REFORMER is more challenging than SLIDE. From an information retrieval's perspective, in attention settings, each data vector is also a query. This unique setting increases the hardness for learning better hash functions.

## F    Efficient GPU implementation

As noted by (Chen et al., 2020), each input in one batch samples a different set of weights(neurons). These irregular operations significantly reduce the benefits of using GPUs: they correspond to different sizes of rows for each input, which makes the forward pass much more difficult to parallelize and thus, more challenging to map to GPUs than regular programs (Burtscher et al., 2012). For this reason, (Chen et al., 2020) build their SLIDE system in C++ from scratch on CPU. Even though their implementation achieved remarkable speed up, their impact is limited as they implemented their system from scratch in C++, making it difficult for the community to adopt SLIDE in practice. To verify that such randomized algorithm is not GPU friendly, we implement the exact same algorithm for GPUs. We show in Section 4.1.1 that it indeed fails to gain any speed up.

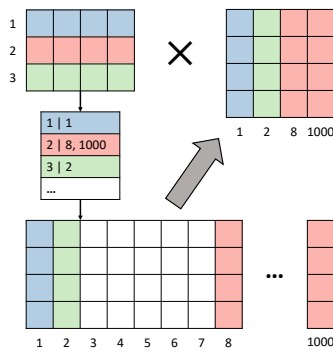

Figure 14: Visualization of how union selection within the batch instead of independent sets of neurons with variant length can avoid irregular memory access.

We design a variant of the algorithm to exploit fast matrix multiplications on GPUs. In the original SLIDE, each training example in a batch retrieves its own subset of weights. Here, we take a union of the retrieved subsets in each batch to avoid irregular and unbalanced memory access, shown in figure 14. Our implementation of this proposal is written in python under Pytorch framework with Cython compiled LSH and CUDA kernels for hashcode efficient computation. We believe this implementation would be more beneficial for the community as it can be easily plugged into any deep learning models.

**Practical Implementation:** In our design, we borrow insights from the theoretical guarantee from above and aim to find the sweet spot between theory limitations and practical challenges. First, instead of detecting weight changes at the cost of an extra copy, we reduce the problem to detect low quality of retrieved weights. In NN training setting, we measure quality as the inner product and low quality indicates a low inner product. We argue these two approaches are similar that they both signal the necessity of updating the data structure. Besides, rather than a soft margin for detecting data changes, which can be ambiguous under dynamic setting, inner product is a better measurement as it clearly indicates the performance of current data structure. More importantly from an efficiency perspective, quality detection comes almost for free because the inner product between the query embedding and retrieved neurons are necessary for the forward pass.

## G    Broader Impact

With the exponentially growing data volume, the scale of neural network (NN) models keeps increasing to achieve unprecedented performance on tasks in many domains, such as recommendation systems (He & Chua, 2017; Medini et al., 2019), natural language processing (Bengio et al., 2003; Vaswani et al., 2017) and computer vision (Deng et al., 2009; Vaswani et al., 2017). However, training those large-scale models imposes challenges on the computational and memory resources even with advanced modern hardware. For example, the recent GPT-3 model (Brown et al., 2020) has a capacity of 175 billion parameters. Therefore, there is an emergent need to reduce the cost of training those giant models.

We now quantify the potential impacts of our work. MONGOOSE sheds lights on efficient NNS-ds for efficient training of deep neural networks. Especially, the slow change observation along with the smart scheduler demonstrate the possibility of applying NNS-ds with larger indexing overhead for dynamic similarity search where the distribution of data changes slowly. We show applications of MONGOOSE on language modeling and recommendation tasks. But since our key or fundamental observations are general, MONGOOSE could potentially generalize to more deep learning models and tasks. On the other hand, more NNS-ds equipped with our observations and scheduler could be involved in the deep learning community to tackle the efficiency issue.

