# OpenReview forum: "MONGOOSE: A Learnable LSH Framework for Efficient Neural Network Training"
_ICLR.cc/2021/Conference — ICLR 2021 Oral_

### Official Review · AnonReviewer3 · 2020-10-18
**Enabling LSH in a learnable framework to speed up neural network training is quite interesting and critical in deep learning era. Experiments on recommendation and language modeling tasks verify the effectiveness of MONGOOSE. Besides, the related formal analysis also seems solid.**

**Rating:** 8
**Confidence:** 4

**Review:**

+++Pros.
-----The observation that model parameters evolve slowly is quite inspiring for more efficient neural network training.
-----The paper proposes MONGOOSE, which is equipped with a scheduler to adaptively perform LSH updates and learnable hash functions to improve query efficiency.
-----Experiments demonstrates the effectiveness of the proposed method, and ablation studies give the readers further insights.

+++Cons.
-----The paper is overall good, but with some minors, such as “Figure 5 shows P@1 and P@5 for MONGOOSE and other baselines during the training process.” in Section 4.1.1.
-----Besides, there are several mathematical symbols should be explained clearly when they first appeared, such as “w” in definition 2.1, “C1, C2” in assumption 3.1, “t_r” in assumption 3.2.

+++Conclusion.
-----Based on the above analysis, I would prefer to make an “ACCEPT” recommendation.
-----By the way, I’m curious about why you named your method “MONGOOSE”? Could you give some reasons?

+++Suggestions.
-----Better make the mathematical symbols more clearly for readers.

---

> ### Author Response · Authors · 2020-11-20
> **Reply to Review by AnonReviewer3**
>
> We thank the reviewer for the strong support of our work! We have addressed definition and typo issues in the updated paper.
>
> **Q: By the way, I’m curious about why you named your method “MONGOOSE”? Could you give some reasons?**
>
> **Response**: We name our framework MONGOOSE from the story Rikki-tikki-tavi (https://www.cs.cmu.edu/~mongoose/rtt.html): “A **FULL** meal makes a **SLOW** mongoose, and if wanted all his strength and quickness ready, he must keep himself thin.” Our framework is **SPARSE** and **FAST**. Thanks for asking! We are very proud of this name and hope you like it.

---

### Official Review · AnonReviewer4 · 2020-10-28
**Good intuition and novel method, but fair practical performance**

**Rating:** 7
**Confidence:** 4

**Review:**

Summary:
The authors make a good insight into the slowly changing of Locality-Sensitive Hashing (LSH) hash codes for the weights (or model parameters) during the Neural Network (NN) training. With this new insight, they introduce a framework Mongoose with a newly designed schedule mechanism to reduce the LSH update overhead. The authors also analyse their model and show some bounds for batch speedup and LSH maintenance. Experimental results validate the efficiency and effectiveness of Mongoose over original LSH methods in NN training.

Pros:
1. First of all, I like the problem the authors focus on. Efficient NN training is very necessary. The idea of using LSH for acceleration is a good direction.
2. The observation of the slow change of LSH hash codes is good and impressive. The authors also conduct many experiments to validate this observation.
3. The idea of using a scheduler for lazy updating is interesting, and they add theoretical analysis about it. And based on the experiments, it seems that this scheduler can capture the changing of model parameters.
4. The proposed framework Mongoose seems to be general for different deep learning model.

Cons:
1. Compared to the good insight and the promising framework, the practical improvement is fair. Although the authors provide many experiments to demonstrate the effectiveness of Mongoose, I still suggest the authors conduct more experiments about parameter studies and memory usage which may be good to improve the paper quality. More details can be found in minor comments (1 & 2) later.
2. Since the idea of the scheduler is inspired by a former work (Cohen et al., 2019), I suggest the authors add a discussion about their connection and difference.
3. The presentation is not very clear. Many notations are used without pre-defined. More details can also be found in minor comments (3 ~ 6) later.
4. For the learnable LSH (Section 3.3.1), when selecting positive/negative samples, the authors use the inner product, while for the loss function, they consider cosine similarity. I suggest the authors make an illustration about the use of these two different measures.

Minor Comments or Typos:
1. The framework Mongoose consists of many parameters. A fresh user may do not know how to set up them. A discussion of parameter settings or some experiments about parameters studies seems to be necessary.
2. The authors claim Mongoose is memory-efficient for NN training. However, I cannot find analysis about space overhead or experiments about memory usage. It should be done to correspond to this claim.
3. For Assumption 3.1, what are the definitions of C_1 and C_2? Is user-defined or auto determined? Since Theorem 3.3 and some Lemma in the appendix also use these two notations, are they the same? For a rigorous expression, I think some declarations are necessary. Also, in Theorem 3.3, the condition of r for g_r is not clear.
4. More problems can be found in Algorithm 1. It seems the core idea of this paper, but there are many typos and errors. For example,
The parameter ‘A’ is not defined and may be unnecessary in Initialize function (line 2).
In line 4, what is the definition of \epsilon_{mp}? Does it have any connection with \epsilon_{mds}? In order to have guarantee for LSH, c_1 - \epsilon_{mp} should be larger than c_2 + \epsilon_{mp}. How to ensure this?
Why setting 1.5 for a smooth cut (lines 13-15)? Is it an empirical value or has any benefit?
In line 17, LSH may update with w^{new}_{\pi([r])} instead of \pi([r]).
5. The y-axis of Figure 3 should be \Delta H (left-subfigure) and \Delta W (right-subfigure) instead of Hamming and L2 distance.
6. The caption of Figure 5 is not clear. And for the sentence “Figure 5 shows P@1 and P@5 for Mongoose and other baselines…” in the paragraph of results in Section 4.1.1, “and P@5” should be removed.

In summary, I like the motivation and the observation in this paper, and the method Mongoose looks good, but I still have many concerns about the idea. So at this stage, I first give a borderline for this paper. I hope the authors can address my concerns.

====================================================================================================

Update: Thank you for your new experiments and detailed feedback. Most of the concerns have been addressed and the experimental results look better. I believe this paper will provide new insight into efficient neural network training. Thus, I raise my rating and recommend this paper to be accepted.

---

> ### Author Response · Authors · 2020-11-20
> **Reply to Review by AnonReviewer4**
>
> We thank the reviewer for the constructive and detailed suggestions. We appreciate your generous comments on the “impressive” observation and “novel” method.
>
> **Q1.1: Compared to the good insight and the promising framework, the practical improvement is fair.**
>
> Response: We have added a closer analysis of the speedup and memory savings of the linear layer where MONGOOSE is applied during training in Section 4.1.1 and Table 2. It shows that the speedup is up to 20$\times$, and memory is 4.5$\times$ in Table 2 compared to 3$\times$ (time) and 1.5$\times$ (mem) in Table 1 in the original version. We want to clarify that the main result in Table 1 is based on MONGOOSE's end-to-end performance (the whole model) during training. We have noticed this presentation does not show MONGOOSE's actual gain and benefit. With this additional analysis, we can separate the effect of an earlier large embedding layer. (The efficient embedding layers is an independent interest and has been explored in [3,4], which we do not aim to accelerate in this work.) We sincerely thank the reviewer for pointing this out so that we can improve the way of presenting our results.
>
> **Q1.2:  Although the authors provide many experiments to demonstrate the effectiveness of Mongoose, I still suggest the authors conduct more experiments about parameter studies and memory usage which may be good to improve the paper quality. More details can be found in minor comments (1 & 2) later.**
>
> Response:  We deeply agree that conducting a more in-depth ablation and sensitivity study on MONGOOSE’s parameters and memory usage can lead to better practicality and usability of our framework for wider audiences. Therefore, we have added:
> 1. the memory usage in Table 1, Table 2, and Section 4.1.2 along with more detailed comparisons and discussions in Appendix D. Specifically, Table 2 shows MONGOOSE has up to 4.5$\times$ memory reduction when isolating the first embedding layer where it does not operate in extreme classification tasks. Table 1 shows that it has a 1.5$\times$ reduction. Section 4.1.2 presents MONGOOSE has up to 6$\times$ memory reduction in language modeling tasks.
> 2. the ablation study of parameters of the scheduler in Section 4.2 and also learnable LSH in Section 4.3. In general, most of the parameters are robust, except for the number of hash functions of LSH. Specifically, for the scheduler, recall that in Algorithm 1, $\epsilon_{mds}$ is the key parameter because the number of neurons exceeding this threshold determines if LSH updates are needed. In practice, instead of choosing a single $\epsilon_{mds}$, we use some fraction of the $\ell_2$ norm of the weights as the update threshold. That fraction is robust to any choice of the fraction in the range 0.1-0.3. For learnable LSH, we found it is not sensitive to the threshold of positive and negative pairs selection but a good choice of the number of hashes would maximize the advantage of MONGOOSE.
>
>
> **Q2: Since the idea of the scheduler is inspired by a former work (Cohen et al., 2019), I suggest the authors add a discussion about their connection and difference.**
>
> Response: The discussion about MONGOOSE scheduler and Cohen et al., 2019 [1] is added in Appendix B. (Cohen et al. 2019) originally designed the data structure for dynamic matrix inversion maintenance, and we adopt it for dynamic LSH maintenance problems. Given the underlying problem is completely different, it requires several non-trivial generalizations and some new ideas. This includes more fine-grained analysis in Lemma B2, B3, B4. Moreover, in our proof, we care about the absolute distance between vectors (i.e., $||w_i - v_i||_2$) rather than the relative difference between real numbers [1], and therefore, we need a new construction of the potential function (see Lemma B8).
>
> **Q3: The presentation is not very clear. Many notations are used without pre-defined. More details can also be found in minor comments (3 ~ 6) later.**
>
> Response: Thanks for pointing out the presentation problem. We have updated our paper to address the notation issues:
> 1. Definition: We add a paragraph summarizing the notations used in Algorithm 1 in Section 3.2.1.
> 2. Consistency: We have changed all $\epsilon_{mp}$ to $\epsilon_{mds}$ and corrected the definition of $g_r$. We also remove redundant variables like A that are no longer used in the paper.
> 3. Clarity: We add a detailed explanation to the $C_1$, $C_2$, $\epsilon_{mds}$, $g_r$ and the remark that only $\epsilon_{mds}$ needs fine-tuning in NN training (see remark 3.4).

---

> > ### Author Response · Authors · 2020-11-20
> > **continued**
> >
> > For your detailed comments.
> >
> > **For Assumption 3.1, what are the definitions of C_1 and C_2? Is user-defined or auto determined? Since Theorem 3.3 and some Lemma in the appendix also use these two notations, are they the same? For a rigorous expression, I think some declarations are necessary. Also, in Theorem 3.3, the condition of r for g_r is not clear.**
> >
> > **Response:** We have added the detailed definition of $C_1$ and $C_2$ in Section 3.1. Briefly speaking, $C_1$ is an upper bound on the (expected) movement of the weights of the neural network, $C_2$ is an upper bound on the variance. They need not be known in advance or fine-tuned (see remark 3.4). We correct the expression of $g_r$.
> >
> > **More problems can be found in Algorithm 1. It seems the core idea of this paper, but there are many typos and errors. For example, The parameter ‘A’ is not defined and may be unnecessary in the Initialize function (line 2). In line 4, what is the definition of \epsilon_{mp}? Does it have any connection with \epsilon_{mds}? In order to have guarantee for LSH, c_1 - \epsilon_{mp} should be larger than c_2 + \epsilon_{mp}. How to ensure this? Why setting 1.5 for a smooth cut (lines 13-15)? Is it an empirical value or has any benefit? In line 17, LSH may update with w^{new}_{\pi([r])} instead of \pi([r]).**
> >
> > **Response:** We correct these typos in Algorithm 1. In particular, (1) eps_{mp} equals eps_{mds}, (2) we remove redundant notation like A, (3) the constant 1.5 is arbitrary, the theoretical property of our scheduler holds as long as it belongs to (1, 2), (4) we take $\epsilon_{mds}$ to be sufficiently small (and greater than $\frac{1}{log^2 n}$) in theory. In practice, we need fine-tune $\epsilon_{mds}$. We also update some discussion on the influence of this parameter (see Section 4.2).
> >
> > **The y-axis of Figure 3 should be \Delta H (left-subfigure) and \Delta W (right-subfigure) instead of Hamming and L2 distance.**
> >
> > **Response:** We correct the caption of Figure 5 and rewrite the sentence.
> >
> > **Q4: For the learnable LSH (Section 3.3.1), when selecting positive/negative samples, the authors use the inner product, while for the loss function, they consider cosine similarity. I suggest the authors make an illustration of the use of these two different measures.**
> >
> > **Response:** Because hashing operates on unit-length vectors, [2] normalizes key vectors: “ we additionally normalize the length of the keys K”. Therefore, selecting samples based on inner product and cosine would be the same. But to avoid confusion, we have added this Section 3.3.1.
> >
> > [1] Cohen, Michael B, et al “Solving linear programs in the current matrix multiplication time” STOC 2019
> >
> > [2] Kitaev et al. "Reformer: The efficient transformer." ICLR 2020.
> >
> > [3] Maxim et al “Deep Learning Recommendation Model for Personalization and Recommendation Systems” Arxiv 2020
> >
> > [4] Xiangyu et al “Memory-efficient Embedding for Recommendations” Arxiv 2020

---

> > > ### Comment · AnonReviewer4 · 2020-11-21
> > > **Reply to authors**
> > >
> > > Thank you for your new experiments and detailed feedback. Most of the concerns have been solved and the experimental results look better. I will raise my rating and suggest to accept this paper.

---

> > > > ### Author Response · Authors · 2020-11-21
> > > > **Thank you!**
> > > >
> > > > Thank you again for helping us improve the paper!

---

### Official Review · AnonReviewer2 · 2020-10-29
**A principled approach to updating LSH based ANNS for faster matrix multiplication in NN training**

**Rating:** 7
**Confidence:** 4

**Review:**

Some neural network runs involve layers with a large number of neurons. These require large matrix-vector or matrix-multiplication which can slow their training/inference. However, if the output of mat-vec/mul is dominated by a few neurons with which the activation has large inner product (a matmul can be thought of as a weighted sum of inner products), then the computation can be sped up by approximating mat-vec/mul by a limited weighted sum with the dominant terms. This requires maintaining an ANNS data structure that is up to data with the back prop. These updates to ANNS have to be done carefully -- too frequent and the training will slow down, or too infrequent and the results of the mat-mul are way off.  This paper studies how to do this in a principled way using data-dependent LSH updates and backs it up with experimental data.

The ideas and algorithms explored in the papers are as follows:
1. The weights changed in a limited manner over time, so it should be possible to take advantage of this.
2. Concentrated changes to a subset of weights can be tracked and patched upon.
3. An LSH update rule to make these changes effectively
4. An earlier algorithm that is reused to decide when the LSH scheme is updated.

The paper also talks about how to identify the layers that benefit the most from this scheme. then it goes on to show the training time benefits of the smart scheduler and the update scheme on extreme classifications tasks as well as transfomers.


Few questions:
1. Why no serious consideration of graph based ANNS? They are data dependent and SOTA for search efficiency and it is possible to update them. Why is LSH the better choice for ANNS here? This needs a rigorous argument.
2. Is this really a general and serious enough problem amongst practitioners that a solution merits publication at a popular conference? It might be, in which case a better quantification of potential impact can help.
3.   Is wiki-325K really the largest dataset for XC? What about larger language models -- sec 4.1.2 seems to study more medium sized networks. Larger scale experiments could make this paper more compelling.

I would more strongly recommend this paper if these questions  can be addressed.

---

> ### Author Response · Authors · 2020-11-20
> **Reply to Review by AnonReviewer2**
>
> We appreciate your concise and precise summarization of our work! We have carefully thought through all your great questions and added corresponding experiments and detailed discussions to answer them in the updated paper. We provide details below:
>
> **Q1: Why no serious consideration of graph based ANNS? They are data dependent and SOTA for search efficiency and it is possible to update them. Why is LSH the better choice for ANNS here? This needs a rigorous argument.**
>
> **Response:** Because our key observation is general, theoretically, our scheduler can generalize to any near-neighbor search data structure, as mentioned in Section 3.2. Our work could open up opportunities for more NNS data structures besides LSH. However, we choose an LSH-based data structure in the current framework because other methods (e.g., graph-based) [1, 2, 3] optimize for fast retrieval speed, but we need a data structure that has a low update or rebuild overhead. LSH is known to have an efficient and simple updating and rebuilding process [9]. We include the discussion of data-dependent indexing in Appendix E.2.
>
> To verify and solidify the above claims, we conduct two additional experiments comparing the updating/rebuilding time of HNSW[1] and our learnable LSH in Section 4.3 (Page 9). Specifically,
> 1. We benchmark the update-time of HNSW and learnable LSH. HNSW is up to 20$\times$ slower at the scale of 300k data points.
> 2. We perform an end-to-end evaluation of using learnable LSH and HNSW in MONGOOSE. HNSW is 5$\times$ slower and 5 point drop in precision@1.
>
> The experimental details are in “Other data-dependent NNS data structures”, Section 4.3.
>
> In conclusion, currently, LSH-based data structure has been chosen in MONGOOSE based on the above reasons. But this motivates us for future work to accelerate the other ANNS data structures or even designing new ANNS data structures. Our framework can also include them to accelerate NN training further.
>
> **Q2: Is this really a general and serious enough problem amongst practitioners that a solution merits publication at a popular conference? It might be, in which case a better quantification of potential impact can help.**
>
> **Response:** Thanks for the great suggestion. We have added a broader impact discussion in Appendix G and extended our conclusion section.
>
> With the exponentially growing data volume, the scale of NN models keeps increasing to achieve unprecedented performance on tasks in many domains, such as recommendation systems [4,5], natural language processing [6], and computer vision [7]. However, training those large-scale models imposes challenges on the computational and memory resources even with advanced modern hardware. For example, the recent GPT-3 model [8] has a capacity of 175 billion parameters. Therefore, there is an emergent need to reduce the cost of training those giant models.
>
> Recent advances in using LSH to break the computational or memory bottleneck in NN training have achieved notable results. However, LSH has been originally explored in the “static” setting where queries are changing, but data is fixed, like approximate near-neighbor search(NNS). Directly applying LSH in the “dynamic” setting where both queries and data are changing has introduced high overhead. Fortunately, in MONGOOSE, we have made a key observation, which provides the opportunity to overcome this overhead in a principal way.
>
> We now quantify the potential impacts of our work. MONGOOSE’s slow change observation and the smart scheduler demonstrate the possibility of applying ANNS data structures with larger indexing overhead for dynamic similarity search where the distribution of data changes slowly.
>
> 1. We have shown applications of MONGOOSE on language modeling and recommendation tasks. But since our key or fundamental observations are general, MONGOOSE could potentially generalize to more deep learning models and tasks.
> 2. On the other hand, more ANNS data structures equipped with our observations and scheduler could be involved in the deep learning community to tackle the efficiency issue. This motivates future work of designing new ANNS data structures or improving the update efficiency of existing ones.
>
>
> **Q3: Is wiki-325K really the largest dataset for XC? What about larger language models -- sec 4.1.2 seems to study more medium-sized networks. Larger scale experiments could make this paper more compelling.**
>
> **Response:** We have updated our main result of the language modeling task (Section 4.1.2) with a 10-layer model (rather than the 6-layer one in the original version) to verify the superiority of MONGOOSE on even larger models.
>
> We hope the above answers the questions and addresses your concerns. Please let us know what other things we could do to improve our paper.

---

> > ### Author Response · Authors · 2020-11-20
> > **References**
> >
> > [1] Gong et al. iDEC: Indexable Distance Estimating Codes for Approximate Nearest Neighbor Search. VLDB 20.
> >
> > [2] Wang et al. Randomized Algorithms Accelerated over CPU-GPU for Ultra-High Dimensional Similarity Search. SIGMOD 18.
> >
> > [3] Deng et al. Pyramid: A General Framework for Distributed Similarity Search.
> >
> > [4] Xiangnan he et al “Neural factorization machines for sparse predictive analytics” SIGIR 2017
> >
> > [5] Medini et al “Extreme Classification in Log Memory using Count-Min Sketch: A Case Study of Amazon Search with 50M Products” NeurIPS 2019
> >
> > [6] Vaswani et al “Attention is all you need” NeurIPS 2017
> >
> > [7] Deng et al “Imagenet: A large-scale hierarchical image database” CVPR 2009
> >
> > [8] Brown et al “Language models are few-shot learners” Arxiv 2020
> >
> > [9] Yiqiu et al “Randomized Algorithms Accelerated over CPU-GPU for Ultra-High Dimensional Similarity Search ” SIGMOD 2020

---

> > ### Comment · AnonReviewer2 · 2020-11-24
> > **Comments on author response**
> >
> > Q1. I agree that rebuilding HNSW from scratch can be slow, and thanks for adding this discussion and measurements. The measurements add weight to the choice of LSH. However, I would still be open to the possibility of updating graph based indices -- this is ongoing work in the community and would not conclude that graph based ANNS is not easy to update.
> >
> > Q2. Thanks for the context. I understand a quantitative summary of impact would be difficult to produce so I am fine with your arguments.
> >
> > Q3. On the XC repo (http://manikvarma.org/downloads/XC/XMLRepository.html), there are datasets much larger than the wiki-325K dataset you have experimented with. Is there a reason you picked this and not the largest? If there is no strong reason to the contrary, please consider presenting results on the largest dataset.

---

> > > ### Author Response · Authors · 2020-11-25
> > > **Reply to Additional Comments by AnonReviewer2**
> > >
> > > **Q1. I agree that rebuilding HNSW from scratch can be slow, and thanks for adding this discussion and measurements. The measurements add weight to the choice of LSH. However, I would still be open to the possibility of updating graph based indices -- this is ongoing work in the community and would not conclude that graph-based ANNS is not easy to update.**
> > >
> > > Yes, we agree that it is an ongoing work to reduce the updating overhead of graph-based NNS-ds in the community, and we will be excited to include it in our framework in the near future. We have revised Section 4.3 to reflect this.
> > >
> > > It would be very helpful to further enrich our related work if you can also point us to any work we have missed on dynamic maintenance of graph-based indices.
> > >
> > > **Q2. Thanks for the context. I understand a quantitative summary of impact would be difficult to produce so I am fine with your arguments.**
> > >
> > > Thanks for your understanding!
> > >
> > > **Q3. On the XC repo (http://manikvarma.org/downloads/XC/XMLRepository.html), there are datasets much larger than the wiki-325K dataset you have experimented with. Is there a reason you picked this and not the largest? If there is no strong reason to the contrary, please consider presenting results on the largest dataset.**
> > >
> > > Thanks for pointing this out! We have been in the process of running larger experiments and focused on the enwik8 experiment for our last revision.
> > >
> > > Amazon-670K and Amazon-3M are the two largest publicly available datasets (Ads-1M and 9M are not). We have added the results for Amazon-670K, which shows 6.5$\times$ speedup (compared to the 3$\times$ on Wiki-325K) in Section 4.1.1. In fact, the empirical results, especially the ones we added based on your suggestions, show that MONGOOSE would have more advantage when the NN layer's size grows because there is more room for MONGOOSE to accelerate. We appreciate your suggestion which helps us better understand the scalability of our algorithm!
> > >
> > > Due to the limited time and resources during the rebuttal, we have added the results for Amazon-670K to show the advantage of MONGOOSE on even larger models. The experiments for Amazon-3M are currently in progress, and we will definitely add the results to the next version when we are allowed to make the edits again.

---

### Official Review · AnonReviewer1 · 2020-10-30
**The framework presented in the paper adds decent contributions, but substantial improvements are needed in writing and presentation.**

**Rating:** 7
**Confidence:** 3

**Review:**

Summary of the paper:

This paper introduces a framework that uses an LSH sketches to improve time an memory bottlenecks neural network training. Specifically, an LSH sketch is used to approximate the matrix multiplications involved in training. It is observed that networks' weights get stable after small number of epochs therefore frequent updates to LSH sketches (which is expensive) are not required. This paper uses data dependent/learnable LSH methods that better adapt to data in order to improve the performance (query and update) of the sketches. Ample experiments are provided to validate the results.

Quality:

Needs improvements in notations. For example, in assumption 3.1, sum is over what? If the indexing is over $i$, then does that means over rows of $w$s (since $w \in \mathbb{R}^{n \times d}$)? Then should the quantities be L2 norms since they are vectors?

Clarity and the presentation can be improved significantly. For example, "Rehash", "Rebuild" functions in algorithm 2 needs to be defined or explained. In general it is better to explain the intuition behind both algorithms 1 and 2.  I believe the figure 1 should clarify the LSH update scheduling, but it is not very clear.

Originality and significance:

I believe that the ideas presented in this paper adds nice contributions to the ICLR community. The idea of using learnable LSH that adapts to the data together with the observation that the weights stabilize after a few epochs is a clever approach to improve the bottlenecks associated with using vanilla LSH sketches.

Other comments:

In regards to the observation with figure 3, I am curious what properties of a dataset leads to this kind of behavior? are there any quantifiable properties? Is this true for any dataset? Are there previous works that explains why this is the case?

Final feedback:
I understand the idea of the proposed framework and the theoretical claims look natural and believable, but the presentation needs to be improved. I am willing to increase my score if the concerns mentioned above are properly addressed.


=====================================================================================================

Added after author response

----------------------------------------

I believe authors have clarified many things I asked and addressed the issues I and other reviewers raised. Therefore, I increase my score. I believe the idea of using LSH for efficient training has a lot of promise and this paper brings a possible way to do this into light.

---

> ### Author Response · Authors · 2020-11-20
> **Reply to Review by AnonReviewer1**
>
> Thanks for your encouragement and suggestions, which have helped us improve the paper!
>
> **Q: Needs improvements in notations. For example, in assumption 3.1, sum is over what? If the indexing is over i, then does that means over rows of ws (since w \in R^{n \times d})? Then should the quantities be L2 norms since they are vectors?**
>
> **Response:** We have updated our paper to address the notation issues. In particular, we add a paragraph summarizing the notations in Section 3.2.1 and add more explanations. For your question, yes, the sum is over rows in assumption 3.1. The assumption is to bound the $L_2$ norm of the weight changes, which corresponds to the observation in Figure 3 above.
>
> **Q: Clarity and the presentation can be improved significantly. For example, "Rehash", "Rebuild" functions in algorithm 2 needs to be defined or explained. In general, it is better to explain the intuition behind both algorithms 1 and 2. I believe the figure 1 should clarify the LSH update scheduling, but it is not very clear.**
>
> **Response:** We have added Figure 8 along with a description of intuition for Algorithm 1 in Appendix B (Page 16) and Figure 9 for Algorithm 2 in Appendix C1 (Page 25). The operation of updating or training LSH hash functions is defined as REHASH, and the operation of updating the hash tables based on the updated hash functions and weights is defined as REBUILD. We have added it to Section 3.3.1.
>
> **Q: In regards to the observation with figure 3, I am curious what properties of a dataset leads to this kind of behavior? are there any quantifiable properties? Is this true for any dataset? Are there previous works that explain why this is the case?**
>
> **Response:** We have added a corresponding interesting discussion about the phenomenon that a sharp drop of weight changes at the early stages of training in Appendix A (Page 15).
>
> This phenomenon can be observed in the text (Transformers), recommendation (embedding models), and image(CNN) datasets, so it is a general one empirically. We have not observed specific dataset properties for this behavior. But several works in the literature conjecture it is related to the optimization algorithm (i.e., SGD), based on empirical observations in CNN training dynamics. For example, [1] conjectures that initially, SGD visits increasingly sharp regions, reaching a maximum sharpness determined by both the learning rate and the batch-size of SGD; [1] infers that it optimizes faster while finding a good sharp region. It is also discussed in a concurrent paper [2], which connects the phenomenon to critical learning periods defined in [3]. Thanks for this profound question, and we have added the related works to support further our key observation of slowly changing hash codes.
>
> We hope the updated draft and the above responses answer your questions. Especially Figure 8 and 9 have made a better presentation of our algorithms. Please let us know if you have more concerns.
>
> [1] Jastrzębski et al. On the relation between the sharpest directions of DNN loss and the SGD step length. ICLR19.
>
> [2] Agarwal et al. Accordion: Adaptive Gradient Communication via Critical Learning Regime Identification.
>
> [3] Achille et al. Critical Learning Periods in Deep Networks. ICLR19.

---

> ### Author Response · Authors · 2020-11-24
> **Thank You!**
>
> We are glad our revision has addressed the concerns and we thank the reviewer for more strongly supporting our paper!

---

### Author Response · Authors · 2020-11-20
**Revision Summary**

We thank all the reviewers for the time and effort in helping us improve the quality of the paper. We were glad that the reviewers found the problem **interesting, necessary and critical** ( R2, R4), the observation  **smart, inspiring and impressive** (R1, R2, R3, R4), and the approach or algorithm **principle, novel and clever** (R1, R2, R3, R4). The reviewers also agreed that the theoretical analysis was **solid and believable** (R1, R3) and the experiments were **ample and effective** (R1, R4).

We have updated the paper to incorporate constructive suggestions. We summarize the major changes:
1. [R4] an analysis of the speedup and memory savings of the linear layer where MONGOOSE is applied during training in Section 4.1.1 and Table 2.
2. [R4] the memory usage in Table 1 and Section 4.1.2 along with more detailed comparisons and discussions in Appendix D.
3. [R4] an ablation study of parameters of the scheduler in Section 4.2 and also learnable LSH in Section 4.3.
4. [R2] a comparison between the updating time of HNSW (a graph-based ANNS data structure) and our learnable LSH in Section 4.3.
5. [R2] updated our main result of the extreme classification task with an additional larger dataset in Section 4.1.1 and the language modeling task with a 10-layer model in Section 4.1.2.
6. [R1, R2, R3, R4] a notation section in Section 3.2.1 and addressed all other notation concerns in Section 3.
7. [R4] a discussion on the connection between the MONGOOSE scheduler and a former work (Cohen et al., 2019) in Appendix B.
8. [R2] a broader impact discussion in Appendix G.
9. [R1] illustrations for Algorithm 1 in Figure 8 (Appendix B.1) and Algorithm 2 in Figure 9 (Appendix C.1) along with descriptions.

---

### Decision · Program_Chairs · 2021-01-07
**Final Decision**

**Decision:**

Accept (Oral)

**Comment:**

Thanks for your submission to ICLR.

When the initial reviews were written, three of the four reviewers were positive about the paper.  Everyone felt it was overall a solid contribution, but there were some concerns about the clarity and presentation, as well as some suggestions for additional experiments.  During the rebuttal/response period, the authors did a very nice job in responding to the concerns of the reviewers.  Ultimately, all of the reviews were in agreement after discussion that the paper is strong and ready for publication.   I also like this paper a lot, and find it to be a nice way to combine LSH with NN training.  I am happy to recommend this paper for publication.